



# Upward transport into and within the Asian monsoon anticyclone as inferred from StratoClim trace gas observations

Marc von Hobe[1], Felix Ploeger[1,5], Paul Konopka[1], Corinna Kloss[1,2], Alexey Ulanowski[3], Vladimir Yushkov[3], Fabrizio Ravegnani[4], C. Michael Volk[5], Laura L. Pan[6], Shawn B. Honomichl[6], Simone Tilmes[6], Douglas E. Kinnison[6], Rolando R. Garcia[6], Jonathon S. Wright[7]

[1]Institute for Energy and Climate Research (IEK-7), Forschungszentrum Jülich GmbH, 52425 Jülich, Germany
[2]Laboratoire de Physique et Chimie de l'Environnement et de l'Espace (LPC2E), Université d'Orléans, CNRS, Orléans, France
[3]Central Aerological Observatory, Dolgoprudnyi, Moscow region, Russia
[4]CNR Institute of Atmospheric Science and Climate (ISAC), I-40129 Bologna, Italy
[5]University of Wuppertal, Institute for Atmospheric and Environmental Research, Wuppertal, Germany
[6]National Center for Atmospheric Research (NCAR), POB 3000, Boulder, CO, 80307, USA
[7]Department of Earth System Science, Tsinghua University, Beijing 100084, China

*Correspondence to*: Marc von Hobe (m.von.hobe@fz-juelich.de)

**Abstract.** Every year during the Asian summer monsoon season from about mid-June to early September, a stable anticyclonic circulation system forms over the Himalayans. This Asian summer monsoon (ASM) anticyclone has been shown to promote transport of air into the stratosphere from the Asian troposphere, which contains large amounts of anthropogenic pollutants. Essential details of Asian monsoon transport, such as the exact time scales of vertical transport, the role of convection in cross-tropopause exchange, and the main location and level of export from the confined anticyclone to the stratosphere are still not fully resolved. Recent airborne observations from campaigns near the ASM anticyclone edge and centre in 2016 and 2017 respectively show a steady decrease in carbon monoxide (CO) and increase in ozone ($O_3$) with height starting from tropospheric values of 80–100 ppb CO and 30–50 ppb $O_3$ at about 365 K potential temperature. CO mixing ratios reach stratospheric background values of ~20 ppb at about 420 K and do not show a significant vertical gradient at higher levels, while ozone continues to increase throughout the altitude range of the aircraft measurements. Nitrous oxide ($N_2O$) remains at or only marginally below its 2017 tropospheric mixing ratio of 326 ppb up to about 400 K, which is above the local tropopause. A decline in $N_2O$ mixing ratios that indicates a significant contribution of stratospheric air is only visible above this level. Based on our observations, we draw the following picture of vertical transport and confinement in the ASM anticyclone: rapid convective uplift transports air to near 16 km in altitude, corresponding to potential temperatures up to about 370 K. Although this main convective outflow layer extends above the level of zero radiative heating (LZRH), our observations of CO concentration show little to no evidence of convection actually penetrating the tropopause. Rather, further ascent occurs more slowly, consistent with isentropic vertical velocities of 0.3 - 0.8 K day[-1]. For gases not subject to





microphysical processes, neither the lapse rate tropopause (LRT) around 380 K nor the cold point tropopause (CPT) around 390 K marks the strong discontinuity of the key tracers (CO, $O_3$, and $N_2O$). Up to about 10 to 20 K above the CPT, isolation of air inside the ASM anticyclone prevents significant in-mixing of stratospheric air. The observed changes in CO and $O_3$

likely result from *in-situ* chemical processing. Above about 420 K, mixing processes become more significant and the air inside the anticyclone is exported vertically and horizontally into the surrounding stratosphere.

## 1 Introduction

The Asian summer monsoon (ASM) anticyclone is the dominant large-scale circulation system in the NH summertime upper troposphere and lower stratosphere (UTLS) (e.g. Hoskins and Rodwell, 1995). Deep ASM convection drives vertical transport

of boundary layer air to the UTLS, where the confinement of anticyclonic flow facilitates a persistent chemical signature that has been detected by multiple satellite sensors (Filipiak et al., 2005; Park et al., 2008; Park et al., 2007; Randel and Park, 2006; Randel et al., 2010; Santee et al., 2017; Thomason and Vernier, 2013; Vernier et al., 2011). Randel et al. (2010) illustrated this troposphere-to-stratosphere transport using HCN, a tracer of anthropogenic pollution and biomass burning measured by ACE-FTS, and described the ASM anticyclone as a gateway for boundary layer air from one of the most polluted

areas on Earth to enter the global stratosphere while bypassing the tropical tropopause.

These satellite observations, although effective in demonstrating the seasonal-average signature, do not have sufficient space-time resolution to clarify some of the key characteristics of this transport pathway. There are a number of outstanding questions regarding the role of ASM transport in connecting Asian boundary layer emissions and regional pollution to global tropospheric and stratospheric chemistry, and to the ensuing perturbations in regional and global climate. These questions

include the most efficient vertical transport locations and time scales, the dynamical processes driving these transport patterns, the respective roles of deep convection and large-scale radiatively-balanced ascent, the dominant source regions for air masses that feed into the anticyclone, the extent of dynamical confinement within the anticyclone, the vertical and horizontal transport pathways of air masses exiting the anticyclone, and the quantitative impact of the ASM on stratospheric water vapour, ozone, and aerosols. These questions have been the focus of a large number of studies using chemical transport models,

Lagrangian trajectory models, reanalysis products, and observational data (e.g. Bergman et al., 2013; Fu et al., 2006; Lau et al., 2018; Li et al., 2005; Pan et al., 2016; Park et al., 2009; Ploeger et al., 2015; Ploeger et al., 2017; Vogel et al., 2015; Vogel et al., 2019; Yan et al., 2019; Yu et al., 2017; Nützel et al., 2019).

Of particular interest to this study is a characteristic description of the ASM transport structure that has emerged from recent modelling studies (Bergman et al., 2013; Pan et al., 2016):





I.  Although the chemical signature of uplifted tropospheric species fills the entire anticyclone in the seasonal-average view, deep vertical transport to the anticyclone level occurs primarily in a region in the southeastern quadrant of the anticyclone (consistent with the linearized model for off-equatorial convective forcing of tropical waves presented by Gill, 1980), centred near the southern flank of the Tibetan Plateau and including the southern slope of the Himalayas, northeast India and Nepal, and the northern portion of the Bay of Bengal.

II.  In this preferred uplift region, transport from the boundary layer to the tropopause level behaves like a "chimney", dominated by rapid ascent.

    III.  This behaviour changes at the UT level, where sub-seasonal dynamics drive east-west oscillations of the anticyclone. These 10–to–20–day oscillations mix the uplifted boundary layer air within the large-scale anticyclone, and contribute to mixing of anticyclone air with the background, thus providing a pathway for this air to enter the global stratosphere.

The transport behaviour described in II has been supported by an analysis using MERRA-2 assimilated trace gases and aerosols (Lau et al., 2018), where the peak monsoon-season aerosol transport is described as a "double-stem-chimney cloud" structure that extends from the boundary layer to around 16 km, near the tropopause level. The "double-stem" rapid convective uprising is identified over two localized areas: the Himalayas-Gangetic Plain and the Sichuan Basin in southwestern China.

This rapid uplift is of special interest for ASM research because the relatively short transport time scale makes this pathway an effective route for very-short-lived (VSL) ozone depleting substances (ODS) to reach the stratosphere. The level at which convectively-driven transport transitions to wave-driven slow ascent is among the outstanding issues that must be addressed to further characterize this transport pathway and its effective time scale. Earlier studies often put this level at approximately 360 K potential temperature (e.g. Park et al., 2009) based on average tropical conditions. New satellite observations over the last decade indicate that the ASM region contains significantly deeper convection, indicated by higher frequencies of convective cloud tops above 380 K potential temperature (Ueyama et al., 2018).

Verification of the chimney-like behaviour suggested by model results requires detailed analysis of high-resolution airborne measurements. The StratoClim campaign using the stratospheric research aircraft M55-Geophysica is the first airborne campaign to provide data suitable for such verification. Based out of Kathmandu, Nepal, StratoClim conducted eight re-search flights in the central region of the ASM anticyclone in summer 2017.

In this work, we use airborne in-situ measurements of carbon monoxide (CO), ozone ($O_3$), and nitrous oxide ($N_2O$) collected both inside and outside the ASM anticyclone in the vicinity of the "chimney" regions. We focus on these three trace gases because they provide complementary information on transport, mixing, and processing. CO is a tropospheric tracer that is





enhanced in polluted air and experiences photochemical removal in the UTLS with a lifetime around 2 to 3 months, and is thus a suitable tracer to investigate the vertical reach of deep convection. $O_3$ is photochemically produced in the UTLS and above at rates as large as a few ppb per day. Concentrations of $O_3$ can reach ppm levels in the stratosphere but do not typically exceed a few tens of ppb in the troposphere, making it useful for examining the transition from tropospheric air to stratospheric air. $N_2O$ is a tropospheric tracer with a significantly longer photochemical lifetime than CO, so that the two tracers can be used together to infer transport time scales. $N_2O$ is removed only after substantial residence time within the stratosphere. Significant reductions in $N_2O$ mixing ratios thus indicate in-mixing of aged stratospheric air.

Using these three trace gases together with dynamical variables, we address the following questions:

Q1 Does the vertical distribution of CO and $O_3$ support the occurrence of rapid convective transport up to the tropopause level?

Q2 At what potential temperature level does the time scale of transport change based on observed vertical gradients in the trace gas profiles?

Q3 Where is this transition region located relative to the tropopause?

Q4 At what level do we begin to see significant signatures of mixing with stratospheric air?

The flights and measurements are described in Section 2, which also introduces model tools and specific dynamic coordinates used in the analysis. Statistical analysis of the observations in terms of different horizontal and vertical coordinates and a comparison of observed and simulated $O_3$ vs. CO tracer correlations are presented in Section 3, followed by detailed discussion of the results in the context of the four questions listed above in Section 4. In the concluding Section 5, we draw a summary picture of vertical transport and confinement inside the ASM anticyclone and compare this to other recent studies.

## 2 Methods

### 2.1 Field campaigns

Two airborne field campaigns utilizing the high-altitude aircraft M55 Geophysica were carried out under the umbrella of the EU project StratoClim. Three flights were conducted from Kalamata, Greece, between 30 August and 6 September 2016, and eight flights were conducted from Kathmandu, Nepal, between 27 July and 10 August 2017. Our study mainly adopts a statistical approach to analyse the entire observational data set comprising measurements from all flights. All flight tracks and their positions relative to the typical ASM anticyclone area are shown in Figure 1. A comprehensive campaign overview and descriptions of the purpose and specific meteorological situation for each flight are given by Stroh et al. (2020).



## 2.2 Measurements, dynamic coordinates and analysis tools

### 2.2.1 Carbon Monoxide (CO)

Carbon monoxide was measured by the new AMICA (Airborne Mid Infrared Cavity enhanced Absorption spectrometer) instrument deployed for the first time during StratoClim. AMICA consists of a power module and a pressurized enclosure containing all major optical components and data acquisition hardware. The instrument is placed underneath a dome-shaped structure on top of the Geophysica aircraft, drawing air from a rear facing inlet through a 2-m length of SilcoNert® coated stainless steel tubing. It employs the ICOS (Integrated Cavity Output Spectroscopy, O'Keefe et al., 1999) technique to measure the trace gases Carbonyl Sulfide (OCS), Carbon Dioxide ($CO_2$), water vapour ($H_2O$) and CO in the wavenumber range 2050.25 – 2051.1 $cm^{-1}$. Full instrumental details will be given in a forthcoming paper (Kloss et al., 2020).

CO mixing ratios were retrieved from observed infrared spectra using a transition at 2050.90 $cm^{-1}$ with line parameters taken from the HITRAN 2012 database (Rothman et al., 2013) and no further calibration parameters. The accuracy was cross-checked for a range of mixing ratios (30 – 5000 ppb) prepared from a $5 \pm 0.05$ ppm CO standard (AirProducts) by dilution with nitrogen or argon (argon was used at the lowest mixing ratios because the nitrogen gas bottle contained a CO impurity at a concentration of ~30 ppb). Taking into account uncertainty in the standard and uncertainties in the MFC flows used to dilute it, the overall accuracy is estimated to be better than 5%. For our analysis, we use AMICA CO data at 10-second time resolution. These data have a 1-sigma precision of ~20 ppb, determined mainly by electrical noise in the observed spectra. Individual CO profiles from all flights are provided in Figure S1 in the supplementary material.

### 2.2.2 Ozone ($O_3$)

$O_3$ was measured with 8 % precision at 1 Hz time resolution by the FOZAN-II (Fast Ozone Analyzer) instrument developed and operated by the Central Aerological Observatory, Russia, and Institute of Atmospheric Science and Climate, Italy (Ulanovsky et al., 2001; Yushkov et al., 1999). FOZAN-II is a two-channel solid state chemiluminescent instrument featuring a sensor based on Coumarin 307 dye on a cellulose-acetate-based substrate, and is equipped with a high accuracy ozone generator for periodic calibration of each channel every 15 minutes during the flight ensuring an accuracy better than 10 ppb in the observations. The measured concentration range is 10 – 500 μg $m^{-3}$; operating temperature range is -95 to +40°C; and the operating pressure range is 1100 – 30 mbar (about 0 – 22 km). The instrument was calibrated at the ground before and after each flight by means of ozone generator and reference UV-absorption $O_3$ monitor (Dasibi 1008-PC).

Ozone data are available from two of the three Kalamata flights in 2016 and from six of the eight Kathmandu flights in 2017. Individual $O_3$ profiles from all flights are provided in Figure S2 in the supplement.



### 2.2.3 Nitrous Oxide (N₂O)

N₂O was measured at 90 s time resolution with an average precision of 0.5% and average accuracy of about 0.6% by the High Altitude Gas Analyzer (HAGAR). The instrument operated by the University of Wuppertal comprises a 2-channel gas chromatograph with electron capture detection (ECD) measuring a suite of long-lived tracers (N₂O, CFC-11, CFC-12, Halon-1211, CH₄, SF₆, H₂) and a non-dispersive IR absorption sensor for fast $CO_2$ measurements (Homan et al., 2010). The instrument is calibrated every 7.5 min during flight with either of two standard gases, which are inter-calibrated in the labora-
tory with standards provided by NOAA/GMD. N₂O data are available for all flights of the campaign (note that the first flight on August 30, 2016 in Kalamata, suffers from comparatively poor N₂O precision of ~2%).

N₂O is fairly well-mixed in the troposphere with a global mean surface mole fraction in 2017 of 329.8 ppb; its tropospheric distribution exhibits a steady growth of 0.9 ppb $a^{-1}$, as well as seasonal variation, interhemispheric difference and other geographic variations (on the order of 1 ppb) due to surface sources (Dlugokendky et al., 2018;
https://www.esrl.noaa.gov/gmd/hats/combined/N2O.html). N₂O is destroyed in the mid stratosphere mainly above 25 km by UV light and reaction with O($^1$D), with slow vertical transport to this sink region resulting in a long atmospheric lifetime of 123 (104-152) years (SPARC, 2013). For an air parcel in the lower stratosphere, a N₂O mixing ratio below the tropospheric value thus indicates the presence of a fraction of photochemically aged air that has passed the sink region above.

### 2.2.4 Geolocation and meteorological data

Temperature was measured at 10 Hz resolution with an accuracy of 0.5 K and a precision of 0.1 K by the commercial Rosemount probe instrument TDC. Pressure and geolocation data were obtained at 1 Hz resolution from the Geophysica's avionic system. Potential temperature along the flight track was calculated from these data. Physical properties on larger scales were derived from ECMWF ERA-Interim data (Dee et al., 2011), including wind speed and direction, potential vorticity (PV), vertical velocities and radiative heating, and local temperature profiles to determine the heights of the lapse rate tropopause
(LRT) and cold point tropopause (CPT).

The vertical resolution of ERA-Interim data in the tropopause region is about 1 km, with relevant levels at 132.76, 113.42, 95.98, 80.40, 66.62 hPa (corresponding to about 14.23, 15.33, 16.50, 17.74, 19.05 km). Related to this vertical resolution is an uncertainty in the determination of the tropopause level. The space-time variability of the tropopause over the campaign region and period will be assessed from its standard deviation in the following.




### 2.3 Dynamic Coordinates

#### 2.3.1 Potential temperature relative to the tropopause

Key question Q3 (Section 1) addresses how the ASM vertical transport relates to the tropopause. Based on the temperature profile (note that the tropopause is sometimes also defined in terms of other parameters such as PV or chemical tracers), the

tropopause in the ASM region can be defined either as the CPT, i.e. the altitude of the coldest temperature, or as the LRT, i.e. "*the lowest level at which the lapse rate decreases to 2 K/km or less, provided also the average lapse rate between this level and all higher levels within 2 km does not exceed 2 K/km*" (WMO, 1957). LRT and CPT were linearly interpolated to each point along a given flight track from the reanalysis grid points around. Under tropical conditions, in the majority of the cases the LRT and CPT are found at the same level, i.e., the LRT is also CPT. When they are not co-located, by definition

the CPT is always higher than the LRT, and their separation typically increases with latitude (Munchak and Pan, 2014, see also Figure S3 in the supplementary material that shows the LRT – CPT separation along the StratoClim flight tracks as a function of latitude).

Because both the LRT and the CPT can vary substantially in altitude, potential temperature, and PV, we determine the respective difference in potential temperature units for our observations and introduce this difference as a new vertical coordi-

nate (e.g. Hoor et al., 2004; Ploeger et al., 2017). Pan et al. (2018) have shown that the LRT better identifies the transition from the troposphere to the stratosphere in the tropics, and relative coordinates with respect to the LRT are used in the main figures and discussions below. Corresponding figures with respect to the CPT are provided in the supplementary material to facilitate comparison with studies having a stronger focus on the CPT. For example, Brunamonti et al. (2018) used the CPT to mark the top of an "Asian tropopause transition layer (ATTL)".

#### 2.3.2 Monsoon equivalent latitude

Different meteorological variables have been proposed in the literature to identify the core of the ASM circulation, including geopotential height (Bergman et al., 2013; Randel and Park, 2006), Montgomery stream function (Santee et al., 2017) and PV (Garny and Randel, 2013; Ploeger et al., 2015). All of these approaches give meaningful (and similar) results when applied to monthly mean fields. Here, we follow the approach of Ploeger et al. (2015) based on the maximum PV gradient on

an isentropic surface with respect to a monsoon-centred equivalent latitude (see below). As PV is an approximately conserved quantity it correlates better with tracer distributions that involve small-scale structures, an advantageous quality when applied to high-resolution in-situ measurements.

Monsoon equivalent latitude (MeqLat) was introduced by Ploeger et al. (2015, please refer to this reference for a comprehensive description of the MeqLat concept and its derivation) as a means of describing the position of an air mass relative to



the centre of the ASM anticyclone. On the 380 K potential temperature surface, the location of the absolute PV minimum is defined as the ASM centre corresponding to 90° MeqLat. As one moves away from this "ASM pole", MeqLat decreases as PV increases. The area enclosed by a given PV contour determines the corresponding MeqLat value, analogous to the decrease in latitude when moving away from the Earth's north or south pole in the definition of equivalent latitude as proposed by Nash et al. (1996). The anticyclone border (i.e. the transport barrier separating air masses inside and outside) is character-

ized by a maximum of the PV-gradient with respect to MeqLat and typically lies around 65° MeqLat, so that MeqLat > 65° can be considered as an indicator of air inside the ASM anticyclone (Ploeger et al., 2015). This boundary identification based on the PV gradient only works well in a shallow layer around 380 K. Therefore we use the 380 K MeqLat value at the horizontal location of each measurement to determine whether it was made inside or outside the ASM anticyclone. By definition, this criterion is exact only for observations at 380 K and becomes increasingly uncertain with vertical distance above or be-

low this level. However, it is the best we can do based on PV, and as the focus of our analysis is on the tropopause level (i.e. around 380 K) the impacts of this uncertainty are unlikely to be large. Figure S4 in the supplementary material shows maps of ASM anticyclone frequency for 2016 and 2017 analogous to that shown for 2011 by Ploeger et al. (2015).

## 2.4 Whole Atmosphere Community Climate Model (WACCM)

The Whole-Atmosphere Community Climate Model, version 6 (WACCM6, Gettelman et al., 2019) is used in this study to

provide large-scale dynamical and chemical background for the StratoClim campaign period. WACCM6 is the atmospheric component of the Community Earth System Model Version 2 (CESM2, Danabasoglu et al., 2020; Emmons et al., 2020). The WACCM6 domain extends from the Earth's surface to the lower thermosphere. For the simulation used in this study, the model uses a 0.9° x 1.2° longitude-latitude grid, with 110 vertical levels on a hybrid-pressure vertical grid with a top at about 150 km (Garcia and Richter, 2019). For pressures < 100 hPa, the vertical coordinate is isobaric; at higher pressures the coor-

dinate is hybrid, transitioning to pure terrain following at the surface. The vertical resolution in the UTLS is ~ 0.5 km. WACCM6 uses comprehensive troposphere, stratosphere, mesosphere and lower thermosphere chemistry (TSMLT, Gettelman et al., 2019). Anthropogenic emissions are from the global CAMS (Copernicus Atmosphere Monitoring Service) emission data set version 4, downloaded from the ECCARD data page: www.igacproject.org/sites/default/files/2018-03/Issue_61_FebMar_2018.pdf. Fire emissions are based on the FINN inventory Version 1.5 (Wiedinmeyer et al., 2011).

The WACCM6 simulation used in this study has been performed with observed sea-surface temperature and sea-ice conditions. Atmospheric winds and temperatures are nudged towards NASA GMAO GEOS5.12 meteorological analysis with a Newtonian relaxation of 50 hours below 50 km using a smooth transition to no nudging at higher model levels from 50 to 60 km. The main effect of nudging is to provide meteorological conditions that are consistent with analysed winds and temperature, allowing comparisons between WACCM6 and observed chemical distributions.





## 3 Observations

### 3.1 General overview

Figure 2 gives a full 3-dimensional view of the AMICA CO observations made during both StratoClim campaigns. The data have been averaged into $1° \times 1° \times 1$ km longitude/latitude/altitude grid cells, and then meridional and zonal averages have been projected into longitude-altitude and latitude-altitude space. In the Kalamata region, CO mixing ratios in the free troposphere were typically between 50 and 80 ppb and only reached 100 ppb in the lowermost 3 km, with the latter likely due to upward mixing of local boundary layer air. By contrast, measurements in the free troposphere over Nepal and surrounding regions indicate elevated levels of CO around 100 ppb at altitudes up to about 16 km, in some cases 17 km, i.e. throughout most of the free troposphere. These relatively large CO mixing ratios indicate fast and efficient transport – most likely by convection – from the boundary layer up to this level (see discussion below). Note that the convective activity was weak to moderate during the first half of the 2017 Kathmandu campaign phase and strongly increased in the second half, coinciding with a cooling of the ASM anticyclone and a corresponding rise of θ isentropes in terms of altitude (Stroh et al., 2020, and references therein). For a large-scale perspective of the ASM region during the Kathmandu 2017 campaign, animations of WACCM-simulated CO distributions are provided as a supplement (https://doi.org/10.5446/48163).

At the time of the 2016 campaign, Kalamata was located almost exactly at the border between the tropics and extratropics, defined here as the latitude where the LRT drops sharply from 16 – 18 km in the tropics to 10 – 13 km at higher latitudes. As a consequence, air masses in the tropical UTLS and in the extratropical stratosphere were probed in the same altitude range (15 – 20 km), often during the same flight. Air masses sampled in the extratropical stratosphere show a clear signature of aged stratospheric air, with small CO mixing ratios in the 10 to 20 ppb range. These air masses were not sampled in the Kathmandu area, where the LRT was always located above 16 km altitude and 369 K potential temperature.

### 3.2 Trace gas distributions

In Figure 3, campaign-averaged CO, $O_3$ and $N_2O$ mixing ratios are shown against different horizontal and vertical coordinates. In the top row, where data are shown in the latitude – potential temperature space, average LRT and CPT levels are also shown for different latitudes. Both levels are highly variable, especially in the Kalamata region. To account for this variability when separating mixing ratio data into tropospheric and stratospheric regimes, the middle row of Figure 3 adopts θ units relative to the LRT as new vertical coordinate (defined in Section 2.3.1). A similar figure using the difference in theta relative to the CPT is provided in the supplement (Figure S6). The clear separation between the Kalamata and Kathmandu flights in latitude space becomes less pronounced in the bottom row, where the latitude coordinate is replaced by MeqLat (see Section 2.4.2) to better represent measurement locations relative to the ASM (with MeqLat = 90° being, by definition, at





the centre of the ASM anticyclone). Nevertheless, observations obtained during the Kathmandu flights represent most of the
data at larger MeqLat values.

Based on Ploeger et al. (2015), we use 65° MeqLat as the boundary between inside the ASMA and outside the ASMA. This
must be regarded as an approximation, because this boundary is not always sharp and readily located and can also vary over
time (Ploeger et al., 2015). Moreover, defining the boundary on the 380 K isentropic surface introduces ambiguities for
measurements collected at significantly higher or lower levels (see Section 2.3.2). Inside the ASM anticyclone, we find some
apparent variability in the region around and directly above the LRT, where $O_3$ decreases and $N_2O$ increases toward the cen-
tre of the ASMA. This variability indicates that in-mixing of stratospheric air is more important near the ASM anticyclone
edge than near the centre.

To further investigate the vertical structure inside the ASM anticyclone, layer-normalized frequency distributions of the three
trace gases for all observations collected inside the anticyclone (i.e. with MeqLat > 65 °) are shown in Figure 4 for the verti-
cal coordinates θ (top panels) and Δθ relative to the LRT (bottom panels; a version with potential temperature relative to the
CPT is provided in Supplementary Figure S7). Tropospheric CO mixing ratios of around 100 ppb, a clear signature of pol-
luted air, stretch upward from the PBL through the lower troposphere up to about 370 K potential temperature corresponding
to about 10 – 20 K below the LRT. The mean LRT during the campaign period is located at about 380 K, with maximum
values up to 396 K and minimum values down to 369 K (see Fig. 4). The mean CPT level is located about 10 K higher. Note
that potential temperature is highly perturbed in this active convective region and that there is a particularly sharp vertical
gradient in theta around the LRT, with a 10 K theta difference corresponding to only a few hundred meters in altitude.
Above the LRT, CO mixing ratios > 100 ppb were not observed in these measurements. Above the 360 – 370 K level, CO
gradually decreases with altitude until reaching stratospheric equilibrium values of 25 ppb or lower between 420 and 435 K.
The observed CO decrease with increasing theta is consistent with the expected local photochemical removal during slow
ascent (green shading in the top left panel of Figure 4).

Ozone, as a stratospheric tracer, behaves opposite to CO. Below about 365 K, $O_3$ mixing ratios did not exceed 100 ppb, con-
sistent with expectations for tropospheric air. A significant increase in $O_3$ with increasing altitude was observed above this
level. As with the decrease in CO, this increase in $O_3$ is largely consistent with local photochemical production during slow
ascent, although not all $O_3$ mixing ratios strictly fall into the expected concentration range (green shaded region in the top
middle panel of Figure 4). Exceptions tend to have values characteristic of tropospheric air (i.e., lower than expected $O_3$) and
may thus be explained by slower production, more rapid ascent or some combination of these. A noteworthy feature in Fig-
ure 4 is that the gradual decrease in CO and the gradual increase in $O_3$ show no obvious discontinuities, on average, at the



LRT or CPT. However, during the StratoClim campaign the LRT marked the vertical limit above which even individual incidences of CO > 100 ppb were not observed anymore (Figure 4, bottom left panel).

$N_2O$ measurements are used to assess the role of in-mixing of stratospheric air at different levels. The local tropospheric $N_2O$ mixing ratio is determined by averaging all observations below 360 K potential temperature, which yields a value of 332.7 ± 1.7 ppb. We then examine the decrease in $N_2O$ with increasing altitude above this level (Figures 3 and 4). Significant reductions in $N_2O$ mixing ratios (unambiguously indicating significant stratospheric in-mixing) were observed only above about 395 K potential temperature and for Δθ more than 15 K above the mean LRT. Between 400 and 410 K, these reductions be-

come more substantial and a clear decline with increasing potential temperature is visible above this level. From the $N_2O$ measurements, we can estimate the fraction of aged extratropical air entrained into the anticyclone at 400 K, following the approach of Homan et al. (2010; there estimating the extratropical fraction of TTL air). Defining 'aged' here as residing in the stratosphere longer than 3 months (approximately the lifetime of CO), we select as 'aged' parcels those with CO < 37 ppb (~1/e times the tropospheric CO value of 100 ppb) from the 2016 Kalamata campaign outside the anticyclone, which

exhibit a mean $N_2O$ mixing ratio of 320 ± 3.7 ppb (1 standard deviation) in the potential temperature range 390 to 400K. Allowing for 1 ppb increase between 2016 and 2017 we thus estimate $N_2O_{aged}$ < 324.7 ppb, i.e at least 8 ppb below the mean tropospheric value in the Kathmandu region (332.7 ppb). Inside the anticyclone at 400K potential temperature, the average $N_2O$ decreased by only 2.4 ppb from this tropospheric value, consistent with an average fraction of in-mixed aged air of at most 30% (i.e. 2.4 ppb / 8ppb).

**3.3 CO – $O_3$ tracer correlations**

Mixing is an important physical process impacting the transport of chemical constituents and a key mechanism for irreversible stratosphere-troposphere exchange (Gettelman et al., 2011). Signatures of mixing in the tropopause region are frequently observed as "mixing lines" in (non-linear) correlations between tropospheric (like CO) and stratospheric (like $O_3$) tracers (e.g. Marcy et al., 2004; Hoor et al., 2002; Fischer et al., 2000; Hintsa et al., 1998). Thus, mixing layers separating the tropo-

sphere from the overlying stratosphere are marked by relatively high concentrations of both CO and $O_3$. This mixing itself manifests in the CO–$O_3$ phase space as deviations from the pure tropospheric and pure stratospheric branches, which in the absence of mixing tend to form an L-shaped distribution (Pan et al., 2010; Pan et al., 2007). Such idealized L-shaped CO–$O_3$ correlations have been used many times to illustrate near-perfect segregation of the troposphere and stratosphere without any mixing layer in between (e.g. Konopka and Pan, 2012).

Figure 5 shows CO–$O_3$ correlation plots of all coincident CO and $O_3$ observations from the Kathmandu 2017 campaigns (left) and from WACCM (right; CO–$O_3$ mixing ratios in the WACCM distributions are the model outputs from the grid loca-





tions closest to the observations in both time and space). Observed CO–O$_3$ correlations show neither an ideal L-shape nor distinct mixing lines between the tropospheric and stratospheric branches. Rather, there is a smooth curved transition from the tropospheric CO branch to the stratospheric O$_3$ branch with increasing theta, consistent with a transition layer in which

the ascending air undergoes photochemical processing. This interpretation is supported by quasi-coincident N$_2$O observations largely showing near-tropospheric values in the transition zone between the high-CO tropospheric branch and the high-O$_3$ stratospheric branch (see Supplementary Figure S8).

The spatial distribution of the CO–O$_3$ pairs in the transition layer (dashed magenta rectangle in Figure 5) is shown in Figure 6 by using altitude, pressure and potential temperature as the vertical coordinate (from left to right, respectively). From these

distributions, we deduce a vertical extent of the transition layer (Figure 6) with respect to these coordinates of 16 – 19 km, 70 – 115 hPa and 365 – 415 K. Figure 5 also shows that the lower tropospheric end of the transition layer is clearly located below the LRT and the higher stratospheric end is located above the LRT. A more detailed analysis of LRT and CPT locations within the transition layer is given in Figure 7. Observed CO–O$_3$ pairs with transition layer characteristics are found between 15 K below and 35 K above the local LRT (a few very rare outliers are even found up to 45 K above the LRT), with roughly

one third of the pairs found below and two thirds above the LRT. Transition layer CO–O$_3$ pairs are observed between 30 K below and 30 K above the CPT, with roughly two thirds below and one third above the CPT. The points circled in red in Figure 5 (from a flight on 31 July) are likely associated with fresh convective outflow, for which O$_3$ mixing ratios are somewhat elevated due to lightning NO$_x$.

Compact CO–O$_3$ correlations were also observed during the START08 campaign, during which air masses originating from

the tropical tropopause layer (TTL) were sampled in the extratropical UTLS region (Vogel et al., 2011, their Fig. 3). Similar to our observations within the anticyclone, photochemical processing within the observed air masses outweighed mixing effects. They attributed this result to the long transit times associated with air mass transport from tropical convective outflow to the upper part of the TTL, well above the LZRH, which allows time for gradual changes in the CO–O$_3$ correlation. The relatively fast isentropic transport from the upper part of the TTL to the extra-tropics where the START08 flights took

place is less crucial.

Overall, WACCM represents the measured correlations well. The curved part of the correlation representing the transition layer is clearly visible, and the simulation closely matches observations with respect to CO–O$_3$ coordinates as well as corresponding potential temperature levels. The overall correlation is somewhat more compact in the model compared to the observations, most likely a result of model's lack of natural variability from unresolved processes, such as the turbulent mixing

from gravity waves.





## 4 Discussion

We now address questions Q1 – Q4 formulated in Section 1 based on the results presented in Section 3. In each subsection below, we discuss one question individually. A refined overall picture of vertical transport in the ASMA region near the tropopause is then given in Section 5.

### 4.1 Rapid convective transport

Our results provide clear evidence of chimney-like vertical transport from the boundary layer to the tropopause over Nepal and northern India, where almost the entire troposphere (up to at least 360 and often 370 K) shows CO mixing ratios similar to those in the polluted boundary layer. The absence of observed CO mixing ratios close to or above 100 ppb at higher levels indicates that we did not observe any incidences of immediate convective outflow above the tropopause. However, after

mixing with the local background, signatures of higher reaching convection in CO and $O_3$ are expected to be small and may therefore not be apparent, so we cannot exclude the occurrence of rapid convective uplift reaching up to or even above the tropopause based on our observations. Nevertheless, the absence of significant $N_2O$ reductions below 400 K (Section 3.2) and the shape of the CO–$O_3$ correlations with little indication for direct mixing between the tropospheric and stratospheric branches (Section 3.3) reveal that the chemical composition in the 370 – 400 K region cannot be explained by very deep

convection mixing with aged stratospheric air. Rather, overshoots penetrating the tropopause mix into the tropospheric air slowly rising out of the 360 – 370 K main convective outflow layer. Our observations provide clear evidence that this slow upwelling and not overshooting convection is the major pathway for air crossing the tropopause in the ASM anticyclone and that the fast chimney-like transport related to convection does indeed stop below the tropopause as suggested by Pan et al. (2016), at least on a synoptic scale. It should be noted that this result does not contradict the potential significance of over-

shooting convection for species such as $H_2O$ that are subject to microphysical removal at the CPT.

Although our observations support the hypothesis of significant input into the anticyclone from a chimney region centred near the southern flank of the Tibetan Plateau, the campaign did not cover a wide enough region to characterize the relative contributions of convection from other sources based solely on the observations. Recent studies have demonstrated horizontal transport of convective outflow from locations over China into the ASM anticyclone (Lee et al., 2019; Yuan et al., 2019)

as well as injections into the ASM anticyclone from tropical typhoons (Li et al., 2017; Li et al., 2020). Evidence for such import into the air masses probed in 2017 has been shown in trajectory studies (Bucci et al., 2019; Lee et al., 2020).



## 4.2 Transition to slow upwelling

From our observations, we constrain the top of the convective outflow layer to about 16 km altitude or 370 K potential temperature, although a few incidences of convective signatures are observed up to 380 K. Above this level, the gradual decline

in CO and gradual increase in $O_3$ suggest continued slow ascent on timescales comparable to those on which CO is photochemically destroyed and $O_3$ is photochemically produced (Figure 4 ). This dynamically driven slow upwelling is radiatively balanced and is less geographically confined than the convective uplift. Pure trajectory models show an "upward spiralling motion of air" extending over large parts of the ASM anticyclone (Vogel et al., 2019; Legras and Bucci, 2020).

This picture is consistent with mean diabatic heating rates from the ERA-Interim reanalysis in July and August (Figure 8).

Relatively strong total diabatic tendencies exceeding 1 K day$^{-1}$ extend upward into the upper troposphere in the tropical part of the monsoon region, south of ~30°N and east of ~70°E. This diabatic "chimney" results from frequent deep convective activity up to around 370 K, although single convective events may reach higher. Legras and Bucci (2020), show high clouds in 2017 to be mostly distributed between 340 and 370 K, with some rare convective events reaching up to 400 K This is very similar to the cloud top height distribution over the ASM region that was shown for 2007 by Ueyama et al. (2018).

It can be seen in the right panels in Figure 8 that the convective activity extends clearly above the level of zero radiative heating (LZRH). Thus, radiatively-balanced upwelling (Figure 8, middle panels) leads to continued upward motion of the air masses. In ERA-Interim, positive isentropic vertical velocities between 0.5 and 1 K day$^{-1}$ (comparable with upwelling in the "shallow branch" of the Brewer-Dobson Circulation in the tropics) are present well above the 380 K level. The observed CO decline and $O_3$ production with increasing potential temperature (Figure 4) point to somewhat slower upwelling, but this is

not quantitatively conclusive because the respective photochemical processing rates are only rough estimates. It should also be noted that other reanalysis data sets may yield quantitatively different results (e.g. Wright and Fueglistaler, 2013; Tao et al., 2019).

## 4.3 Relationship of the transition region with respect to the tropopause

As stated in Section 4.1, our observations show no evidence of convection crossing the tropopause. At respective mean po-

tential temperature levels of 380 K (minimum: 369 K, maximum: 396 K) and 390 K (min: 370 K, max: 411 K), both the LRT and the CPT are located within the radiative upwelling regime described in Section 4.2. CO mixing ratios in the ASM anticyclone do not drop any more sharply at the LRT or CPT than in the regions immediately above or below these levels, and they remain greater than stratospheric background concentrations up to about 40 K above the LRT and 25 K above the CPT. The absence of any sharp transition implies that neither the LRT nor the CPT represents a vertical transport barrier for

these species (neglecting, for the present discussion, microphysical processes and constituents affected by them). That the





LRT and CPT are located well below the level of significant stratospheric in-mixing (see Section 4.4) implies that these levels do not represent separation between the troposphere and the stratosphere in the ASM anticyclone in either a dynamical or chemical sense. Thus the "tropospheric bubble" (Pan et al., 2016) not only extends to an exceptionally high tropopause but even above that tropopause.

Because the air appears to be largely isolated within the ASM anticyclone as it rises through the transition region, vertical cross-tropopause transport clearly occurs within the ASM region. This vertical transport is reproduced to some degree by the WACCM simulations shown by Pan et al. (2016), where concentrations of both CO (Figures 2 and 7 in Pan et al., 2016) and the E90 tracer (Figure 9 in Pan et al., 2016) remain relatively large above the TP in the ASM region up to ~420 K, in good agreement with our observations.

**4.4 Mixing with the stratosphere**

Taking $N_2O$ mixing ratios significantly below the current tropospheric value as an indicator for the contribution of stratospheric air (Section 3.2), mixing of the rising ASM air with older stratospheric air starts to become clearly visible at about 400 K potential temperature. CO mixing ratios at 415 K and above largely match the stratospheric background value (Figure 4 and Section 3.3), indicating that the tropospheric character of the ASM anticyclone ceases at or below this level. Individual

observations of slightly elevated CO are found up to 435 K where, based on radiative upwelling rates and CO lifetimes, CO photochemical decay to equilibrium values is expected to be complete. A contribution of ASM anticyclone air at even higher potential temperature levels cannot be ruled out based on our observations. The significant isolation of the ASM anticyclone air up to 400 K and the rapid weakening of this isolation in the range between 400 and 435 K shown by our observations is roughly consistent with recent analyses of ASM confinement using trajectory analyses (Brunamonti et al., 2018; Legras and

Bucci, 2020).

**5 Conclusions**

In situ observations of CO, $O_3$ and $N_2O$ were collected during two aircraft campaigns near the edge and near the centre of the ASM anticyclone. CO and $N_2O$ are tropospheric tracers with short and long photochemical lifetimes, respectively, while $O_3$ is a mainly stratospheric tracer. Analysis of these observations helps us to further fill in the emerging picture of vertical

transport in the ASM anticyclone, confirming and extending earlier studies:





- A "fast convective chimney" lifts polluted boundary layer air to the ASM upper troposphere, with the main outflow below 370 K. Evidence of this "chimney" occasionally reaches up to the local LRT but was not observed above this level (Figures 2, 3, 4 and 6).

- Inside the ASM anticyclone, above the level of main convective outflow, upward transport of air continues at slower rates roughly consistent with vertical velocities and heating rates from reanalysis data and on time scales consistent with photochemical removal and production of CO and $O_3$ respectively (Figures 4 and 5). Our results are consistent with the idea of air "spiralling" upward inside the ASMA, as recently described by Vogel et al. (2019).

- Air crosses the tropopause vertically during this radiatively-balanced ascent, with neither the LRT nor the CPT marking sharp discontinuities for gases not affected by microphysics: the "*tropospheric bubble*" (Pan et al., 2016) extends above the tropopause (Figures 4 and 7).

- Below about 400 K, air is to a large extent horizontally isolated within the ASM anticyclone and in-mixing of stratospheric air from outside is not a dominant factor ($N_2O$ in Figure 4).

- There is no evidence for a sharp vertical boundary marking the top of the ASM anticyclone. The isolation starts to weaken at 400 K and the degree of mixing with surrounding stratospheric air smoothly increases towards higher levels. Clear signatures of tropospheric air undergoing slow chemical processing are retained up to at least 415 K.

This picture corresponds well to that proposed in a study by Ploeger et al. (2017) who postulated that vertical cross-tropopause transport dominates inside the ASM anticyclone, followed by quasi-horizontal transport along isentropes above the tropopause into the tropics and into the NH extratropical stratosphere. It is also largely consistent with the recent trajectory analysis presented by Legras and Bucci (2020) and with the ATTL picture proposed by Brunamonti et al. (2018). As stated in Section 4.1, our picture of slow upwelling dominating vertical transport across the tropopause does not exclude the occurrence of overshooting events, so Brunamonti et al.'s interpretation of a shallow layer of enhanced $H_2O$ mixing ratios above the CPT as an indication of overshooting convection crossing the CPT is not in conflict with this picture. Given the ~5 K temperature drop in the ASM anticyclone at the beginning of August described by Brunamonti et al. (2018), the enhanced $H_2O$ could also result from earlier ascent through a warmer CPT. The concept of the Lagrangian cold point rather than the local CPT being the relevant feature in limiting $H_2O$ transport has been described for the TTL by Pan et al. (2019), and it will be interesting to investigate this further with $H_2O$ observations from the two airborne campaigns.



*Data availability*. All data will soon be accessible via the HALO database at https://halo-db.pa.op.dlr.de/mission/101 . In the
meantime, they can be provided by the PIs upon request.

*Author contributions*. M. von Hobe, F. Plöger, P. Konopka and L. Pan led and devised the analyses. M. von Hobe and C.
Kloss provided AMICA CO observations. A. Ulanovsky, V. Yushkov and F. Ravegnani provided FOZAN $O_3$ observations.
C. M. Volk provided HAGAR $N_2O$ observations. D. Kinnison, R. Garcia developed the new WACCM-110L version and S.
Tilmes carried out the simulations for this study. S. Honomichel extracted the $CO/O_3$ correlations from WACCM and pre-
pared all figures related to $CO/O_3$ correlations as well as the video supplement showing the WACCM simulations over the
campaign period. J. Wright calculated heating rates from ERA-Interim and prepared Figure 8. M. von Hobe prepared figures
related to the observations and prepared the manuscript with contributions from all co-authors.

*Competing intersts*. The authors declare that they have no conflict of interest.

**Acknowledgements**

We would like to thank the MDB team operating the M55 Geophysica aircraft for the successful deployments and their sup-
port with instrument integrations and operation as well as provision of avionic data, as well as local airport and ATC staff in
Kalamata and Kathmandu for their support. We also thank Johannes Wintel, Valentin Lauther, Thorben Beckert, Emil Ger-
hardt and Lydia Eppert, who supported HAGAR operations and data analysis, and Nicole Spelten for preparing time syn-
chronized merged files that were used for our analyses. Special thanks to Fred Stroh for the tremendous spadework he put in
to actually make the campaigns and flights in these locations possible, and to Colonel Kharki for diplomatic support in Ne-
pal. Campaign planning and logistics as well as the scientific interpretation was largely covered by the StratoClim project
funded by the European Community's Seventh  Framework  Programme (FP7/2007-2013) under  grant agreement n°
603557. Additional support for this work was obtained from the German Bundesministerium für Bildung und Forschung
(BMBF) under the ROMIC-SPITFIRE project (BMBF-FKZ: 01LG1205), and from a joint research project funded by the
National Natural Science Foundation of China (NSFC project number 41761134097) and the German Research Foundation
(DFG project number 392169209). The GEOS data used in the WACCM6 run have been provided by the Global Modeling
and Assimilation Office (GMAO) at NASA Goddard Space Flight Center through the online data portal in the NASA Center
for Climate Simulation. Felix Ploeger was funded by the Helmholtz Association under grant no. VH-NG-1128 (Helmholtz
Young Inves-tigators Group A–SPECi). Corinna Kloss was partly funded by the Deutsche Forschungsgemeinschaft (DFG,
German Research Foundation) – 409585735.



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





**Figures**

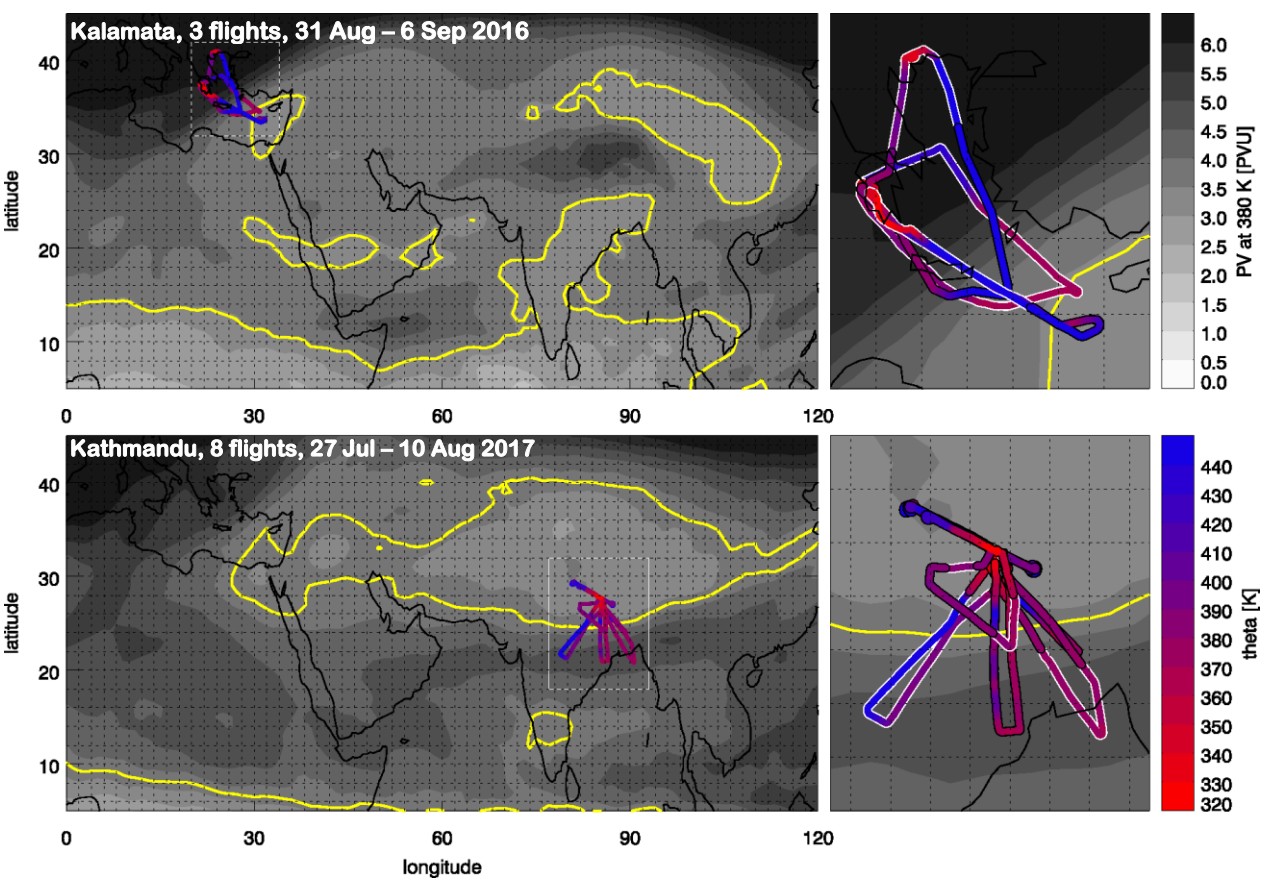


**Figure 1** Maps showing tracks of all Geophysica flights conducted during the StratoClim campaign phases in 2016 (top) and 2017 (bottom) for the larger ASM area (left) and zoomed to the respective campaign area (right). PV contours (grey shadings) are averaged between 29 Aug and 7 Sep 2016 and between 26 Jul and 11 Aug 2017 respectively. Yellow lines represent the average anticyclone boundaries according to the Ploeger et al. (2015) criterion of 3.5 PVU for 2016 and 3.7 PVU for 2017 (maps of the average
ASM anticyclone position according to the Ploeger et al. criterion are shown in Supplementary Figure S4).






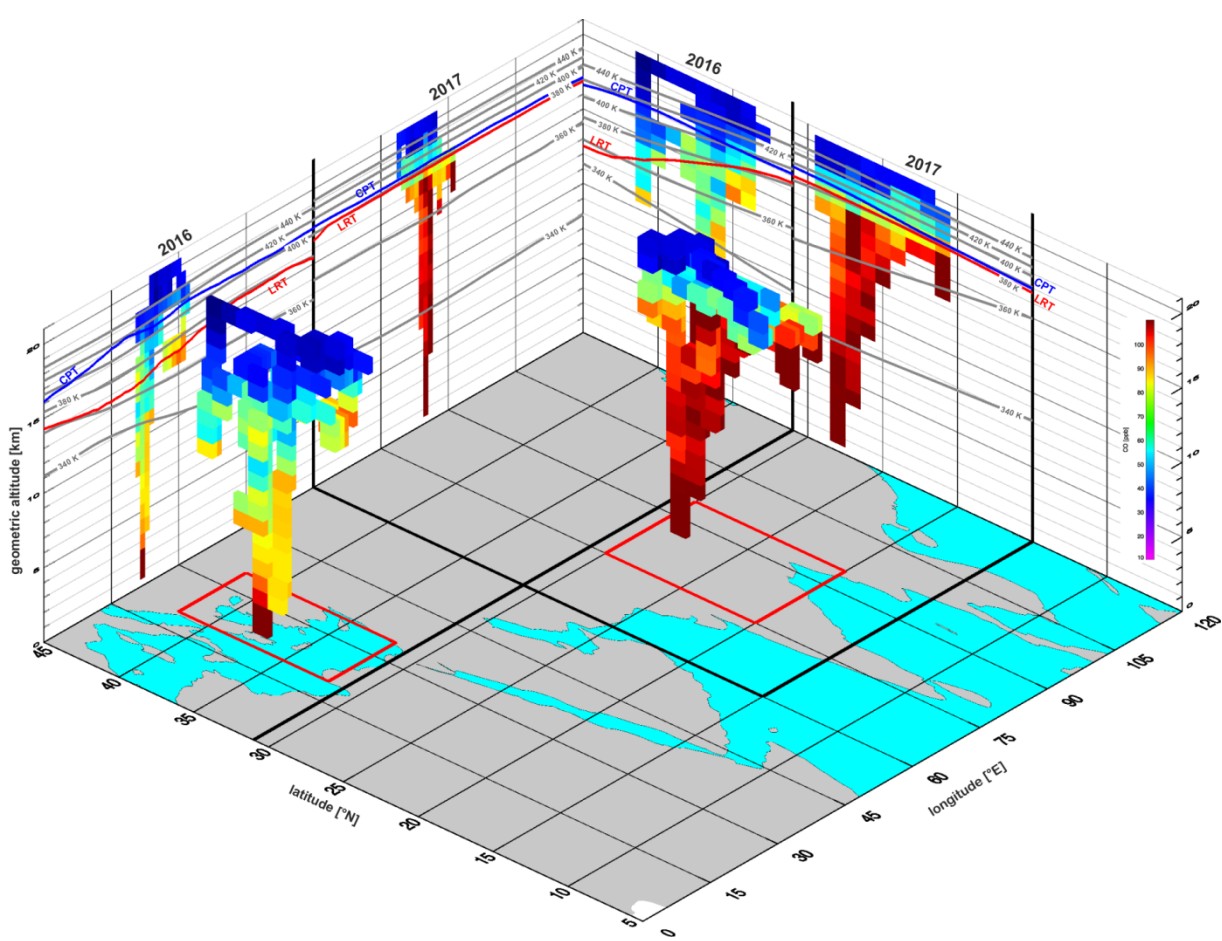

**Figure 2** 3D Map of averaged CO mixing ratios observed during all StratoClim flights (raw CO profiles for all individual flights are shown in supplementary Figure S1). Values in 3D space represent averages in 1° × 1° × 1 km longitude – latitude – altitude bins.
Longitude-altitude averages over all latitudes and latitude-altitude averages over all longitudes are projected onto the x-z and y-z planes respectively. Grey contour lines in the x-z and y-z planes show potential temperature levels averaged meridionally and zonally over the areas marked on the map by the thick black lines; red and blue lines show LRT and CPT heights averaged over the same areas (based on ERA-Interim reanalysis products during the respective campaign phases).





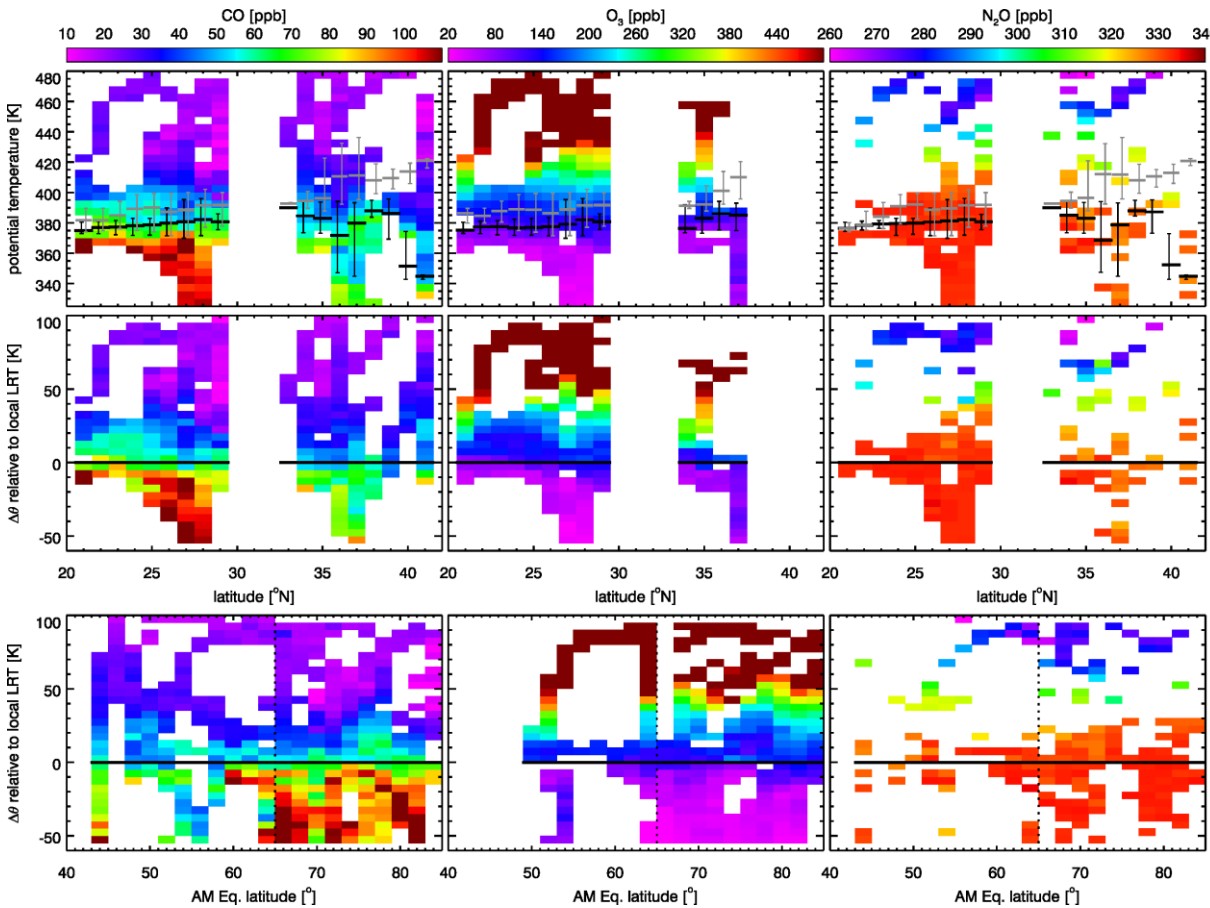

**Figure 3** Top panels show CO (left), O$_3$ (middle) and N$_2$O (right) mixing ratios averaged into 1° latitude × 5 K potential temperature bins. Black and grey bars show average and minimum/maximum LRT and CPT heights respectively for all measurements made in each 1° latitude bin. Note that the bins with valid data for CO, O$_3$ and N$_2$O do not exactly match due to different data coverage for the AMICA, FOZAN and HAGAR instruments. The middle panels show the same data with potential temperature relative to the LRT (see Section 2.3.1) as the vertical coordinate. In the bottom panels, the horizontal coordinate is transformed to MeqLat (see Section 2.4.2). The number of samples and standard deviation in each bin are given in Figure S5 in the supplementary material.





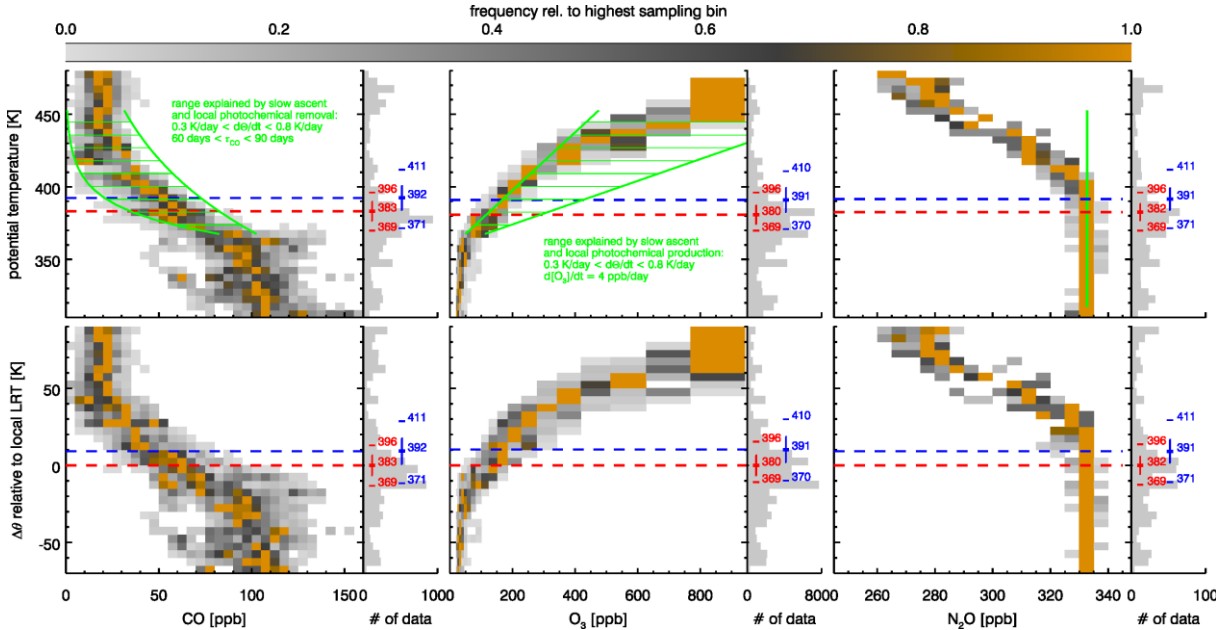

**Figure 4** Layer-normalized relative frequency distributions of CO (left), O$_3$ (middle) and N$_2$O (right) inside the ASM anticyclone (MeqLat > 65 °) for vertical coordinates of potential temperature (top) and potential temperature difference relative to the local LRT (bottom). The number of observations on each vertical level is plotted along the right side of each panel. The mean (with standard deviation), minimum and maximum LRT and CPT levels are also shown in red and blue, respectively. Areas marked in green in the CO and O$_3$ panels indicate the range of concentrations consistent with photochemical removal/production during slow ascent (assuming isentropic ascent rates between 0.3 and 0.8 K day$^{-1}$, based on Ploeger et al., 2010, and Garny and Randel, 2016; a CO photochemical lifetime between 60 and 90 days based on Xiao et al., 2007; and O$_3$ production rates of about 4 ppb day$^{-1}$ based on Fig. 8 in Gottschaldt et al., 2017). In the N$_2$O panel, the green line denotes the average HAGAR N$_2$O in the troposphere (below 360 K potential temperature) during the 2017 campaign.






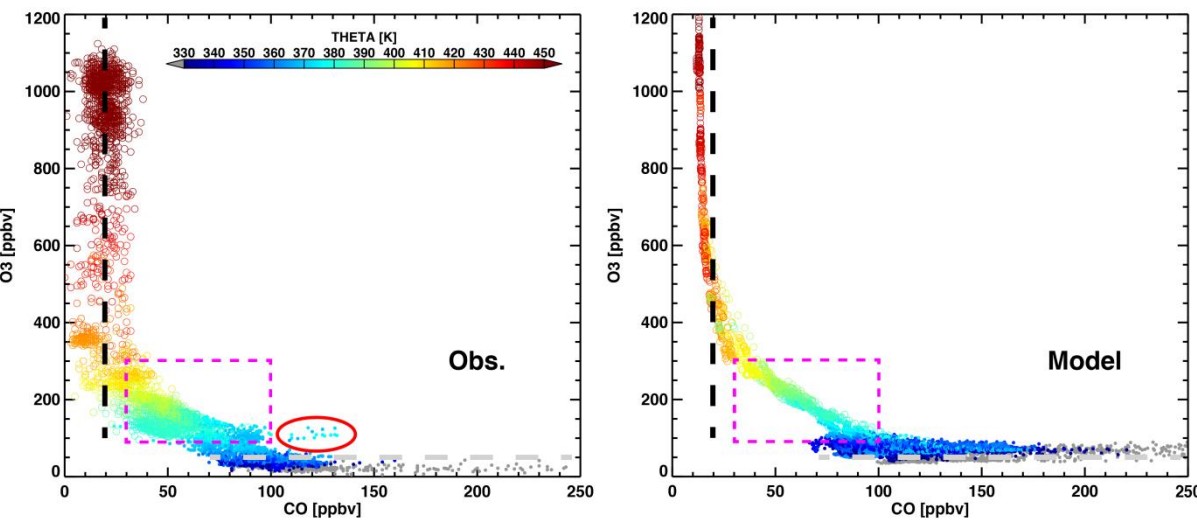

**Figure 5** O$_3$ vs CO relationships in tracer-tracer space based on observations (left) and WACCM simulations (right) for the 2017 Kathmandu flights. The data points are coloured according to their potential temperature level (with grey indicating θ < 330 K) and they are shown by two types of symbols for above (circle) and below (dots) the LRT. WACCM grid points are selected to match the flight dates and are sampled in grid points nearest to the flight track (within ± 1° in latitude and longitude) to minimize space-time
offsets. Dashed lines indicate three "regimes" identified by the O$_3$-CO relationship, corresponding to the troposphere (grey dashed line), the stratosphere (black dashed line) and the transition layer (magenta rectangle). We chose criteria of 30 ppb < CO < 100 ppb and 80 ppb < O$_3$ < 300 ppb to assign measurements to the transition layer (i.e. both gases having neither tropospheric nor stratospheric values). The points marked by the red oval are likely produced by convective transport and O$_3$ production from lightning NOx (see text).






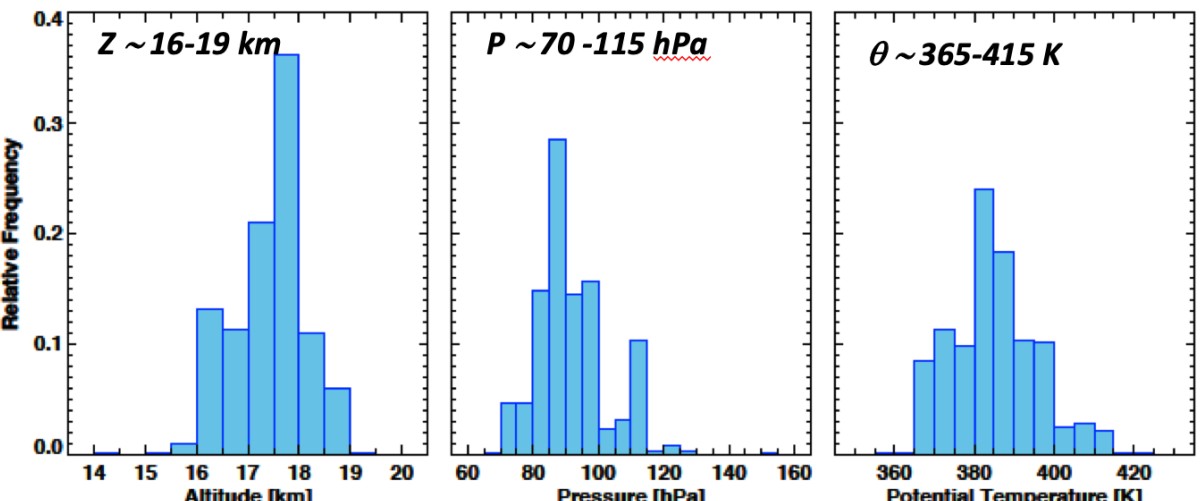

**Figure 6** Relative frequency distributions of transition-layer measurements in altitude, pressure, and θ coordinates.


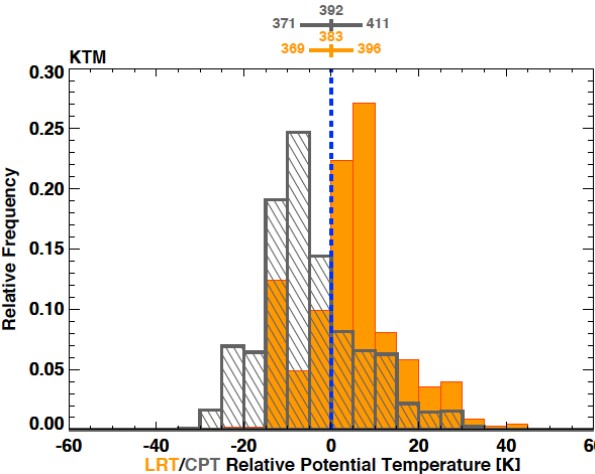

**Figure 7** Transition layer in tropopause-relative potential temperature coordinates. Histograms show normalized relative frequency
distributions of transition layer measurements as defined in the text. The distribution relative to the LRT (CPT) is shown in orange
(grey). The mean and the range of the LRT and CPT in potential temperature coordinates (as shown in Figure 4) are given in match-
ing colours above the zero line.

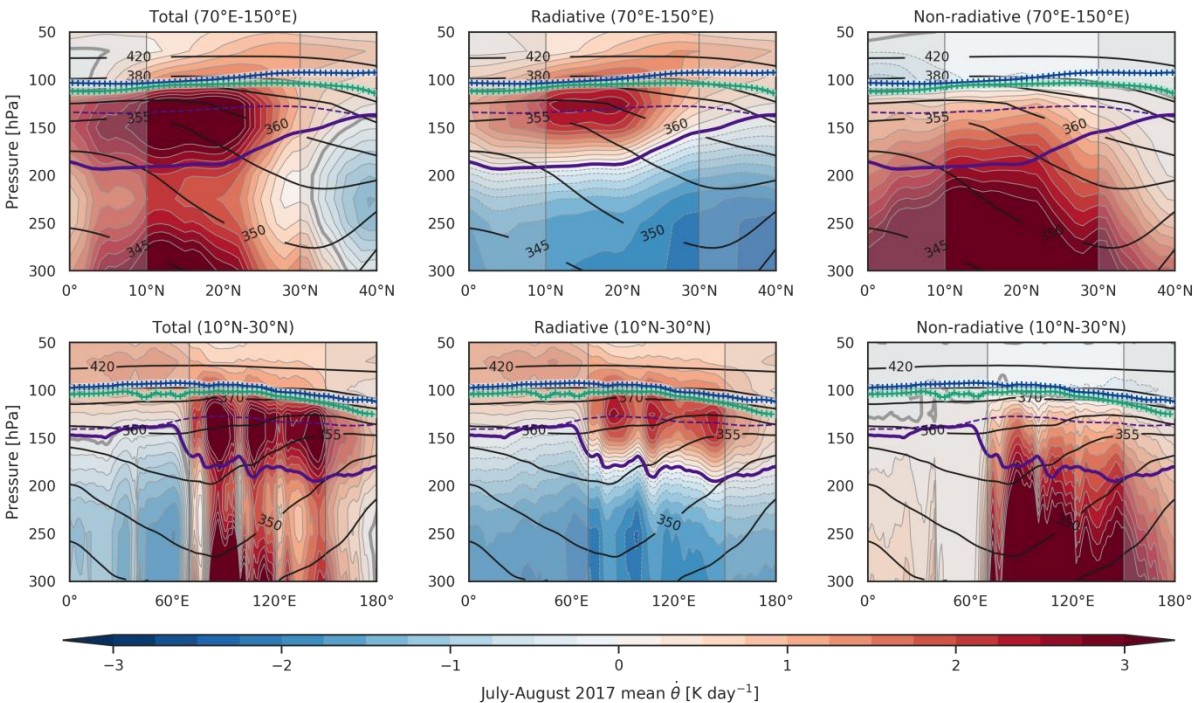

**Figure 8** Zonally (top) and meridionally (bottom) averaged total (left), radiative (centre) and non-radiative (right) diabatic potential temperature tendencies $(d\theta/dt)_{diab}$ based on the ERA-Interim reanalysis for July-August 2017. Zonal averages are calculated over 70 – 150°E and meridional averages over 10 – 30°E with area weights applied. Black contours show potential temperature. Purple contours show the vertical location of the LZRH based on the zero contour in time-mean radiative heating rates under all-sky (solid) and clear-sky (dashed) conditions. LRT and CRT are shown in green and blue respectively. Averaging ranges for θ contours, LZRH, LRT and CPT are the same as those for $(d\theta/dt)_{diab}$.