# Peer review of "Upward transport into and within the Asian monsoon anticyclone as inferred from StratoClim trace gas observations"

_Atmospheric Chemistry and Physics, 2020_

## Referee Comment (RC1) · Michelle Santee (Referee) · 12 Oct 2020

**Review of "Upward transport into and within the Asian monsoon anticyclone as inferred from StratoClim trace gas observations" by von Hobe et al.**

Airborne in situ measurements of CO, $O_3$, and $N_2O$ collected in the Asian summer monsoon (ASM) anticyclone and surrounding regions during the 2016 and 2017 StratoClim field campaigns are analyzed to elucidate troposphere-stratosphere transport pathways and mechanisms. The manuscript is well written, the figures are generally well done, and the supplementary information is helpful and appropriate. I do, however, have a number of comments on both the analysis and the description thereof that I feel should be addressed before the manuscript is accepted for publication.

**Specific comments and questions (major substantive issues and minor points of clarification, wording suggestions, and grammar / typo corrections are listed together for each Section in sequential order through the manuscript):**

**General** (throughout the manuscript)
- In many (perhaps most) cases, acronyms are not spelled out the first time they are used.
- In several places (e.g., L103, L116, L171), "dynamic" should be "dynamical".
- In several places (e.g., L288-289, L354, L374), "incidences" should be "occurrences".

**Abstract**
- L22-25: Are all of the values (both mixing ratios and potential temperatures) quoted in these lines fully consistent with those given in the main text? The tropospheric abundance of $N_2O$ in particular departs from that stated in Section 3.2.
- L31-33: I find the wording of this sentence awkward. One suggestion would be to rewrite along these lines: "For the key tracers (CO, $O_3$, and $N_2O$) in our study, none of which are subject to microphysical processes, neither the lapse rate tropopause (LRT) around 380 K nor the cold point tropopause (CPT) around 390 K marks a strong discontinuity in their profiles."
- L33: It seems a bit odd to focus on the CPT here, when most of the results in the paper are described relative to the LRT (and $H_2O$ is not one of the measurements discussed).

**Introduction**
- L73: "uprising" (which means "revolt" or "rebellion") is not the right word here; I suggest "lofting".

**Section 2**
- Figure 1: I find the description of this figure and its relation to Fig. S4 confusing and the related discussion in the text (one sentence, L113-114) inadequate.
  - Although the color bar label indicates that the contour plots show PV at 380 K, that information should be stated in the figure caption itself. That 380 K is the only isentropic surface on which the method of Ploeger et al. [2015] can be applied is acknowledged later in the manuscript (L206-207), but many readers may not appreciate that limitation when Fig. 1 is introduced.

- The 380 K level is difficult to distinguish in the theta color bar used for the flight tracks. The color palette should be constructed to facilitate identification of the portions of the flights at or near 380 K (for which the defined anticyclone boundary is pertinent). Much of the flight time in both deployments took place at levels well above or below 380 K.
- The overlaid yellow lines in Fig. 1 do not closely resemble the cyan contours in Fig. S4 (especially for the 2016 period), nor do the average PV values quoted in their respective captions match. Is that because Fig. 1 shows the averages over the specific campaign phases, whereas Fig. S4 shows seasonal averages? This point should be clarified.
- L138: Add a comma after "flight".
- L139: The semicolons in this line should be commas.
- L141: of ozone --> of an ozone
- L146: instrument operated by the University of Wuppertal comprises --> instrument, operated by the University of Wuppertal, comprises
- L151: Delete the comma after "Kalamata".
- L154: Dlugokendky --> Dlugokencky
- L157: a $N_2O$ --> an $N_2O$
- L167: Add "and" before the last item in the lists of both pressure and altitude levels.
- L178: the reanalysis grid points around --> the surrounding reanalysis grid points
- L181: material that --> material, which
- L192-193: Quite a number of authors (beyond the short list given here) have used GPH to define the ASM anticyclone boundary; a similar comment can be made about the use of PV, and more than one paper has used MSF as well. Thus, it would be best to add "e.g." in all of these cases.
- L210-211: The discussion of the limitations of the approach used in this study to segregate measurements made inside and outside the anticyclone should explicitly note that the anticyclone varies in size at different levels, and also that it tilts northward with altitude. So while it may be the best that the authors can do, I feel that they are a little too cavalier in dismissing the impact that their approach might have on the interpretation of their results. They state that the focus of this analysis is on the tropopause level near 380 K, but that is not an entirely true statement – in particular, Fig. 4 (based on inside-anticyclone points) spans the domain 310–480 K, and quite a bit of discussion is devoted to the stratosphere above the 400 K level. I do not share their optimism that the inherent ambiguity introduced by their PV-based approach to identifying the anticyclone boundary necessarily has little impact.
- L225-229: I feel that more should be said about the ability of CESM2/WACCM6-SD to faithfully reproduce the observed confinement of trace gases within the ASM anticyclone. Two recent papers [Orbe et al., GRL 2017; ACP, 2020] show the sensitivity of both convection and large-scale circulation to the details of how nudging is implemented. My understanding is that CESM2 includes substantial changes to the nudging scheme and convective parameterizations from CESM1. Has any previously published study demonstrated the fidelity of the model's depiction of the ASM anticyclone for the specific configuration used here? If so, it should be cited. If not, then it would be appropriate to elaborate further on that issue in this paper.

**Section 3**

- L239-240:  Since the Stroh et al. StratoClim overview manuscript is still in preparation, it might be good to add a reference to Bucci et al. [2020] for the point that convection strengthened in the latter part of the Katmandu deployment.  The paper by Bucci et al., which has been accepted for publication in ACP, is already being cited elsewhere.
- Fig. 2 and associated discussion in L247-248:
  - Fig. 2 nicely conveys a lot of information in a compact form that facilitates comparison of the two data sets.  However, the labels on the CO color bar are too small to easily read without substantial enlargement, and the color palette makes it difficult to distinguish between different abundances.
  - It is stated that "small CO mixing ratios in the 10 to 20 ppb range" were measured during the Kalamata flights.  Although it is hard to judge, I do not really see any indication of CO values below about 35–40 ppb, and certainly none as low as 10–20 ppb.
- L248: These air masses --> Such air masses
- L252: in the latitude --> in latitude
- L253: Just to be really clear, I suggest adding "(higher-latitude)" in front of "Kalamata".
- L255: as new --> as the new
- L258: Section 2.4.2 --> Section 2.3.2
- L264-266: This sentence states that at and just above the LRT, $O_3$ decreases and $N_2O$ increases with increasing MeqLat (i.e., "toward the centre of the ASMA").  But to my eye, both appear to change little; if anything, $O_3$ actually increases slightly with MeqLat, whereas $N_2O$ shows only a very slight increase that is barely perceptible in the color scheme used.
- Fig. 3 and its caption:
  - The x-axis labels should be "MeqLat" to match the text.
  - The dotted line marking 65° MeqLat is not defined in the caption.  Moreover, this line is barely visible; a solid (or even a dashed) line would show up better.
  - L711: To ensure clarity, I suggest adding "from both deployments" or something similar in front of "averaged".
  - L716: Section 2.4.2 --> Section 2.3.2
- Fig. 4 and its caption:
  - Obviously, monitors vary, but on my screen the color of green used for the expected range information is too bright to show up well, making these lines hard to read.
  - The x-axis ranges for the panels showing the # of data points lie too close to the mixing ratio panels for both the same species to their left and those of their neighbors to the right, so the individual scales are hard to differentiate in some cases.
  - I assume that the mean, max, and min values of the LRT and CPT vary slightly between the columns because, as explained in the Fig. 3 caption, the bins with valid measurements are different for each instrument, but it would be good to clarify that explicitly in this caption.
  - Is there an explanation for the cluster of samples below 350 K (below –30 K in Δθ) with much lower CO values (~40–70 ppb)?  Some of those bins contain a fair number of points.
  - L721: Since the number of observations is partly obscured by the LRT/CPT ranges, it might be good to add something along the lines of "as the grey histogram" after "plotted".
  - L724: There is no Ploeger et al. [2010] reference – perhaps 2017 is meant?

- o L725: The applied maximum lifetime of UTLS CO of 90 days seems somewhat on the long side to me. Based not only on Xiao et al. [2007], but also Duncan et al. [JGR, 2007] and Holloway et al. [JGR, 2000], I think that the total atmospheric lifetime of CO over Asian continental regions in summer is more like 1–2 months, not 2–3 months. So it is possible that the expected range for CO in this figure should be adjusted.
- L277: Since the fact that no CO mixing ratios exceeding 100 ppb are observed above the LRT is an important point, it might be helpful to add a vertical line on the CO panels in Fig. 4 marking that value.
- L284: In addition to the point that not all $O_3$ mixing ratios fall in the expected concentration range, it is also worth noting that much of the expected range is unpopulated in the data.
- L286: It is suggested that more rapid ascent may account for the apparently tropospheric character of some of the air masses observed to fall outside of the expected range for $O_3$, but wouldn't stronger ascent also affect the CO in those parcels, leading their measured CO concentrations to extend beyond the expected range (which is not seen)?
- L289: Delete "anymore".
- L298: For the reasons mentioned above, I think that this line should be amended to note that 3 months represents the upper end of the range for CO lifetime in the summertime UTLS.
- L301: For clarity, I think it would be good to start this sentence "Considering the upper end of the $N_2O$ range and allowing for a 1 ppb increase between 2016 and 2017, …".
- Fig. 5 and its caption:
  - o Although I have no objection if the authors prefer to leave Fig. S8 in the supplementary material, I note that there is plenty of room in the observational panel of Fig. 5 for a sizeable inset showing the $O_3$-CO relationship color-coded by $N_2O$. The theta color bar could be moved to the model panel or to the space above the two panels (which would also underscore that it pertains to both). In any case, it would help orient the reader to add the magenta box on the correlation plot in Fig. S8.
  - o The grey dashed line in Fig. 5 representing the tropospheric regime is too pale to be easily seen. I think it would be better to make this a black dashed line as well and then add "vertical" and "horizontal" in the caption to distinguish the two black dashed lines.
- L315: campaigns --> flights (or campaigns --> campaign)
- L325-326: It would be appropriate to add tildes in front of all of these ranges, as is done in each panel of Fig. 6.
- Fig. 7: It seems odd to me that the colors used to represent the LRT and CPT have suddenly changed in this figure. It would provide more continuity for the reader to draw the LRT histogram in red and outline the CPT one in blue, in accordance with Figs. 2 and 4. (Of course, the zero line would also need to be colored differently in that case.)
- L328-330: As the authors note, some points lie as much as 45 K above the LRT, thus it is necessary to add "mainly" or some similar qualifier to "are found"; a similar comment can be made regarding the statement about the CPT.
- L331-333: The sentence "The points circled … $NO_x$." feels out of place, tacked on at the end of the paragraph. It might flow better at the end of the previous paragraph. Alternatively, although it is plausible that these points are related to lightning $NO_x$, the authors could do

more to back up that statement (e.g., through references to previous studies), in which case this discussion could make up its own short paragraph, probably at the end of the section.

- L341-345: It is a bit of an understatement to say that the modeled correlation is "somewhat" more compact than that observed – it is considerably more compact, not only in the transition region but also throughout the stratospheric regime. Couldn't the measurement uncertainties (e.g., precision of 20 ppb for CO, 8% for $O_3$) account for some of this scatter?
- L344: of model's --> of the model's

**Section 4**

- L351-353: I find this discussion confusing. The authors state that their results provide clear evidence of vertical transport "to the tropopause", but then they go on to note that the data show transport "up to at least 360 and often 370 K", whereas they have shown that the mean LRT is at 380 K. Moreover, on L374 in the next section, it is stated that convective signatures are occasionally observed up to 380 K. This should be reconciled / clarified.
- L354: The wording in this line is awkward. If I have understood it correctly, then rather than "immediate convective outflow" it would be better to say something along the lines of: "convective outflow above the tropopause immediately prior to the measurement time".
- L355: Here too I think the wording is a bit awkward and unclear. I suggest instead: "mixing with the local background following transport to this level, signatures of deep convection".
- L358-359: It would be better to add commas after "correlations" and "(Section 3.3)".
- L361-365: I would like to see more discussion here placing these conclusions into the context of other recent studies touching on this issue, including Ploeger et al. [2017], Vogel et al. [2019], Yan et al. [2019], and Legras & Bucci [2020] (and possibly others).
- L371: The year for the Bucci et al. reference should be [2020].
- L373: It would be more appropriate to say "the top of the *main* convective outflow layer".
- Fig. 8 and its caption:
  - Again, it would be nice if the colors of the LRT and CPT overlays matched those in previous figures, even though that would mean choosing a new color palette for the contour plots.
  - L758: 30°E --> 30°N
  - L760: CRT --> CPT
  - The caption should explain the shading. One possibility would be to begin the sentence about how the zonal and meridional averages are calculated with "As demarcated by the vertical grey lines and bolder colors on the respective panels, …".
- L382: Delete the comma after "(2020)".
- L387-388: It would be appropriate to include a reference for the speed of the BDC.
- L388-391: It took me a couple of minutes to figure out that the statement that the observed CO decline and $O_3$ production point to upwelling that is somewhat slower than the ERA-I vertical velocities is based on the fact that the green lines in Fig. 4 were derived assuming an ascent rate of 0.3 K day$^{-1}$ < d$\theta$/dt < 0.8 K day$^{-1}$. Since this information appears only in the caption and on the figure itself (not in the main text), it would be good to remind readers of these values here. Moreover, it seems to me that this would also be an appropriate place to remind readers of the uncertainties associated with using PV at 380 K to identify insideanticyclone points, which is another reason that these upwelling estimates are not "quantitatively conclusive", as noted in L390.

- L398-399: As noted in L330-331, Fig. 7 shows the distance above the CPT to be 30 K.
- L399-403: That the tropopause over the ASM region does not represent a vertical transport barrier has been noted in previous studies, e.g., Vogel et al. [2019].
- L405-409: I find this whole paragraph confusing. For one thing, it's not clear whether the first sentence is meant to be a general statement or an expression of the findings of this study, but in any case it should be made clear that this picture has been described in many previous papers. In addition, I'm not sure why the WACCM simulations reported by Pan et al. [2016] are being discussed here, when simulations with a new (presumably improved) version of the model have been run specifically for this study. Can't these points be made with reference to the simulations shown in Fig. 5 and the animation in the supplement?
- L410: Since the analysis here sheds light only on in-mixing of stratospheric air into the anticyclone, whereas most previous studies quantifying the isolation of the anticyclone have focused on parcels escaping from it (and Legras & Bucci [2020] argue that the latter occurs more easily than the former), it might be better to entitle this subsection "In-mixing of stratospheric air".
- L413-417: It would be relevant to note here that the analysis of Vogel et al. [2019] showed transport of young air masses in the ASM circulation up to as high as ~460 K.
- L417-420: The study of Randel & Park [2006] (already cited elsewhere) should probably also be mentioned here, as they performed a trajectory-based quantification of confinement inside the anticyclone over the range 500–70 hPa.

**Conclusions**
- L427: LRT --> LRT around 380 K
- L430: times scales consistent with --> times scales largely consistent with
- L434: The term "tropospheric bubble" was not italicized in L403; usage should be consistent.
- L438-440: These lines are somewhat redundant with the two previous bullet points. In addition, the fact that the degree of mixing with surrounding stratospheric air increases at higher levels was noted by Vogel et al. [2019] as well. Finally, the other bullets point to the specific relevant figures, and it would be good if this one did too.
- L443-444: Unlike Ploeger et al. [2017], Legras & Bucci [2020] found that the "blower" mechanism operates at and above ~360 K, not just above the tropopause. (The studies of Vogel et al. [2019] and Yan et al. [2019] also found substantial horizontal flow out of the anticyclone between 340 and 380 K.) Legras & Bucci further found that localized "chimney"-like behavior ends at the cloud tops, above which ascent follows a broad spiral over the entire anticyclone domain, as shown by Vogel et al. I think it would be beneficial for the authors to put their results into the context of other recent studies in a more detailed and nuanced manner, rather than simply stating that they are "largely consistent".
- L445-449: The importance of extreme deep convective clouds in moistening the ASM region is also emphasized by Ueyama et al. [2018], and Legras & Bucci [2020] also found that penetrative convection may have a significant impact.

**References**
- L490: The paper by Bucci et al. has now been accepted and is in press.
- L570: The paper by Legras & Bucci has now been published.
- L670: hte Comission --> the Commission

**Supplementary Material**
- L771: CRT --> CPT
- L789: on --> in
- L803: Figure 3 --> Figure 4

---

## Referee Comment (RC2) · Anonymous Referee #2 · 17 Nov 2020

Review comments on ACP-2020-891

Upward transport into and within the Asian monsoon anticyclone as inferred from StratoClim trace gas observations

by Hobe et al

The manuscript endeavors to comprehend the field measurements of CO, N2O & O3 from the ASMA region during the StratoClim field campaigns and elucidate its different transport pathways. Quite a lot of quantification and new insights are offered in this manuscript. Hence the manuscript can be accepted for publication after addressing the following points.

[Figure]

Recommendation: Minor revision

Comments/suggetions

1. 3D Map of averaged CO from figure 2, looks nice but hard to get information easily. The Theta_e labels are not visible and difficult to understanding CO mixing ratios difference as they mentioned in the manuscript.

2. Many unconventional acronyms used in the manuscript, which make it difficult to read the manuscript, and some of them are not expanded where it first appeared. It will be better to minimize the same in running text.

3. Hope that AM Eq latitude in the figure 3 label is a typo instead of M Eq Lat. Or Asian Monsoon Equalent latitude is also a better terminology.

4. In case if not much required the supplementary materials can be limited in the manuscript. For eg Figure S4, which is not providing any new insights and besides which creating confusion with the discussion later in the text and main figures.

5. The statements in lines 245's are very difficult to digest from the figure and considering the possibilities of multiple other influencing factors.

6. Why there is a high value of CO even below 340K at designated latitudes (from figure 3 & 4). It will be nice if the authors can provide some details/explanations in this regard.

7. In the discussion section how suddenly authors restricted the transport till 370K. Till this point, the authors were mentioning that the LRT is at 380K. Better to clarify this.

---

## Author Comment (AC1) · 15 Dec 2020

*We thank Michelle Santee for her careful, comprehensive and constructive review of our paper, which helped us to clarify many points that were ambiguously presented, and to greatly improve the analysis and the manuscript.*

*Below, we go through Michelle Santee's review point by point, using* normal font for her original remarks *and* *blue italics* *for our replies.*
* * *
Airborne in situ measurements of CO, $O_3$, and $N_2O$ collected in the Asian summer monsoon (ASM) anticyclone and surrounding regions during the 2016 and 2017 StratoClim field campaigns are analyzed to elucidate troposphere-stratosphere transport pathways and mechanisms. The manuscript is well written, the figures are generally well done, and the supplementary information is helpful and appropriate. I do, however, have a number of comments on both the analysis and the description thereof that I feel should be addressed before the manuscript is accepted for publication. Specific comments and questions (major substantive issues and minor points of clarification, wording suggestions, and grammar / typo corrections are listed together for each Section in sequential order through the manuscript):

**General (throughout the manuscript)**

- In many (perhaps most) cases, acronyms are not spelled out the first time they are used.

*We have carefully checked this and spell out acronyms the first time they are used in the revised version.*

- In several places (e.g., L103, L116, L171), "dynamic" should be "dynamical".

*This has been changed as suggested.*

- In several places (e.g., L288-289,L354,L374), "incidences" should be "occurrences".

*This has been changed as suggested.*

**Abstract**

- L22-25: Are all of the values (both mixing ratios and potential temperatures) quoted in these lines fully consistent with those given in the main text? The tropospheric abundance of $N_2O$ in particular departs from that stated in Section 3.2.

*Numbers given for CO and $N_2O$ in the abstract were indeed incorrect. They had not been updated from an early manuscript version written around figures based on uncalibrated data. This is a clear mistake (that should not have occurred) and will be corrected.*

*For $O_3$, "30 – 50 ppb" in line 22 and "did not exceed 100 ppb" in line 281 (Section 3.2) are both true and not contradictory. But we agree that the statements go into slightly different directions and will add a notion that the bulk tropospheric $O_3$ falls into the 30 – 50 ppb range in Section 3.2.*

- L31-33: I find the wording of this sentence awkward. One suggestion would be to rewrite along these lines: "For the key tracers (CO, $O_3$, and $N_2O$) in our study, none of which are subject to microphysical processes, neither the lapse rate tropopause (LRT) around 380 K nor the cold point tropopause (CPT) around 390 K marks a strong discontinuity in their profiles."

*We adopt the suggested sentence.*

- L33: It seems a bit odd to focus on the CPT here, when most of the results in the paper are described relative to the LRT (and $H_2O$ is not one of the measurements discussed).

*This was also inherited from an early manuscript version, where we had a stronger focus on the CPT rather than the LRT (we obviously should have been more careful when finalizing the abstract just prior to submission).*

*We replace "up to about 10 to 20 K above the CPT" by "up to about 20 to 35 K above the LRT".*

**Introduction**

- L73: "uprising"(which means "revolt "or "rebellion") is not the right word here; I suggest "lofting"

*We change this as suggested.*

**Section 2**

- Figure 1: I find the description of this figure and its relation to Fig. S4 confusing and the related discussion in the text (one sentence, L113-114) inadequate.

  *We expand the discussion of this Figure in the text.*

  o Although the color bar label indicates that the contour plots show PV at 380 K, that information should be stated in the figure caption itself. That 380 K is the only isentropic surface on which the method of Ploeger et al. [2015] can be applied is acknowledged later in the manuscript (L206-207), but many readers may not appreciate that limitation when Fig. 1 is introduced.

  *We make these additions to the revised Figure caption.*

  o The 380 K level is difficult to distinguish in the theta color bar used for the flight tracks. The color palette should be constructed to facilitate identification of the portions of the flights at or near 380 K (for which the defined anticyclone boundary is pertinent).Much of the flight time in both deployments took place at levels well above or below 380 K.

  *The theta color scale is adjusted so that levels close to 380 K can be identified by very pale colors. Further apart from 380 K, colors get deeper, with red for the troposphere and blue for the stratosphere.*

  o The overlaid yellow lines in Fig.1 do not closely resemble the cyan contours in Fig. S4 (especially for the 2016 period), nor do the average PV values quoted in their respective captions match. Is that because Fig. 1 shows the averages over the specific campaign phases, whereas Fig. S4 shows seasonal averages? This point should be clarified.

  *Fig. S4 shows averages for the period 1 July – 31 August, Fig. 1 averages everything (PV contours and Ploeger criterion boundaries) only over the campaign periods. We make this clear in the revised Fig. 1 caption.*

- L138: Add a comma after "flight".

*We add a comma as suggested.*

- L139: The semicolons in this line should be commas.

*We replace the semicolons by commas as suggested.*

- L141: of ozone --> of an ozone

*We change this as suggested.*

- L146: instrument operated by the University of Wuppertal comprises--> instrument, operated by the University of Wuppertal, comprises

*We change this as suggested.*

- L151: Delete the comma after "Kalamata".

*We delete the comma as suggested.*

- L154: Dlugokendky--> Dlugokencky

*We correct as suggested.*

- L157: a $N_2O$ --> an $N_2O$

*We correct as suggested.*

- L167: Add "and" before the last item in the lists of both pressure and altitude levels.

*We add "and" as suggested.*

- L178: the reanalysis grid points around--> the surrounding reanalysis grid points

*We change this as suggested.*

- L181: material that --> material, which

*We change this as suggested.*

- L192-193: Quite a number of authors (beyond the short list given here) have used GPH to define the ASM anticyclone boundary; a similar comment can be made about the use of PV, and more than one paper has used MSF as well. Thus, it would be best to add "e.g." in all of these cases.

*We add "e.g." to these references as suggested.*

- L210-211: The discussion of the limitations of the approach used in this study to segregate measurements made inside and outside the anticyclone should explicitly note that the anticyclone varies in size at different levels, and also that it tilts northward with altitude. So while it may be the best that the authors can do, I feel that they are a little too cavalier in dismissing the impact that their approach might have on the interpretation of their results. They state that the focus of this analysis is on the tropopause level near 380 K, but that is not an entirely true statement – in particular, Fig. 4 (based on inside-anticyclone points) spans the domain 310–480 K, and quite a bit of discussion is devoted to the stratosphere above the 400 K level. I do not share their optimism that the inherent ambiguity introduced by their PV-based approach to identifying the anticyclone boundary necessarily has little impact.

*We phrase this more carefully and point out the potentially larger impact at higher levels.*

- L225-229: I feel that more should be said about the ability of CESM2/WACCM6-SD to faithfully reproduce the observed confinement of trace gases within the ASM anticyclone. Two recent papers [Orbe et al., GRL 2017; ACP, 2020] show the sensitivity of both convection and large-scale circulation to the details of how nudging is implemented. My understanding is that CESM2 includes substantial changes to the nudging scheme and convective parameterizations from CESM1. Has any previously published study demonstrated the fidelity of the model's depiction of the ASM anticyclone for the specific configuration used here? If so, it should be cited. If not, then it would be appropriate to elaborate further on that issue in this paper.

*This is an excellent suggestion. We include a new paragraph elaborating on the performance of the 110 level run in the region of ASM UTLS based on some initial evaluations.*

**Section 3**

- L239-240: Since the Stroh et al. StratoClim overview manuscript is still in preparation, it might be good to add a reference to Bucci et al. [2020] for the point that convection strengthened in

the latter part of the Katmandu deployment. The paper by Bucci et al., which has been accepted for publication in ACP, is already being cited elsewhere.

*We include this reference here as suggested.*

- Fig.2 and associated discussion in L247-248:
  - Fig. 2 nicely conveys a lot of information in a compact form that facilitates comparison of the two data sets. However, the labels on the CO color bar are too small to easily read without substantial enlargement, and the color palette makes it difficult to distinguish between different abundances.

  *We do not want to change the color palette, because it was chosen to be exactly the same as the one used in Figure 3. We make it larger and increase the size of the labels.*

  - It is stated that "small CO mixing ratios in the 10 to 20 ppb range" were measured during the Kalamata flights. Although it is hard to judge, I do not really see any indication of CO values below about 35–40 ppb, and certainly none as low as 10–20 ppb.

  *While the color bar is the same as in Figure 3, the actual 3D plot was accidentally done using a different color palette. This is corrected and the CO abundances should now (i) be easier to distinguish and (ii) match numbers in the text and in Figures 3 and S1.*

- L248: These air masses --> Such air masses

*We change this as suggested.*

- L252: in the latitude --> in latitude

*We change this as suggested.*

- L253: Just to be really clear, I suggest adding "(higher-latitude)"in front of "Kalamata".

*We add this as suggested.*

- L255: as new --> as the new

*We change this as suggested.*

- L258: Section 2.4.2 --> Section 2.3.2

*We correct this.*

- L264-266: This sentence states that at and just above the LRT, $O_3$ decreases and $N_2O$ increases with increasing MeqLat (i.e., "toward the centre of the ASMA"). But to my eye, both appear to change little; if anything,$O_3$ actually increases slightly with MeqLat, whereas $N_2O$ shows only a very slight increase that is barely perceptible in the color scheme used.

*The observation is correct. The sentence was originally written with reference to the CPT (i.e. looking at Figure S6) because CPT was used as the "main tropopause" in an early version of the manuscript. Obviously, the LRT tilts differently with MeqLat, so the observation does not apply to the LRT space shown in Figure 3. Because the statement is not central to our conclusions, we choose to delete it altogether.*

- Fig. 3 and its caption:
  - The x-axis labels should be "MeqLat" to match the text.

  *The x-axis labels are changed as suggested.*

  - The dotted line marking 65° MeqLat is not defined in the caption. Moreover, this line is barely visible; a solid (or even a dashed) line would show up better.

  *We make this line thicker and mention it in the caption.*

- o L711: To ensure clarity, I suggest adding "from both deployments" or something similar in front of "averaged".

*We add this as suggested.*

- o L716: Section 2.4.2 --> Section 2.3.2

*We correct this.*

- Fig. 4 and its caption:

  - o Obviously, monitors vary, but on my screen the color of green used for the expected range information is too bright to show up well, making these lines hard to read.

*We change this color to purple.*

*Note that we also change the range for CO lifetimes (in response to the comment below) and the range for ascent rates (we noticed that the 0.3 – 0.8 K/day$^{-1}$ do not actually reflect the available literature, and this original choice was actually based to some extent on a misinterpretation of heating rate q in terms of isentropic upwelling rate d$\theta$/dt, which is obviously higher). The rationalization for both ranges is extended and therefore moved from the figure caption to the main text.*

  - o The x-axis ranges for the panels showing the # of data points lie too close to the mixing ratio panels for both the same species to their left and those of their neighbors to the right, so the individual scales are hard to differentiate in some cases.

*We remove the "0" at each # panel x-axis and state in the caption that this axis minimum is always zero.*

  - o I assume that the mean, max, and min values of the LRT and CPT vary slightly between the columns because, as explained in the Fig. 3 caption, the bins with valid measurements are different for each instrument, but it would be good to clarify that explicitly in this caption.

*We add an explicit statement in this respect as suggested.*

  - o Is there an explanation for the cluster of samples below 350 K (below –30K in Δθ) with much lower CO values (~40–70 ppb)? Some of those bins contain a fair number of points.

*All these were observed during the Kalamata campaign (cf. Figure S1), mostly below the local LRT and thus reflect the free troposphere over the Mediterranean. This appears to illustrate the problem with defining MeqLat at 380 K fixed, and here it obviously doesn't work at 350 K.*

*We add a sentence in this respect in Section 3.2.*

  - o L721: Since the number of observations is partly obscured by the LRT/CPT ranges, it might be good to add something along the lines of "as the grey histogram" after "plotted".

*We add this as suggested.*

  - o L724: There is no Ploeger et al. [2010] reference –perhaps 2017 is meant?

*There is a Ploeger et al. (2010) reference that was missing from the reference list. But this reference has been removed from the new extended discussion of ascent rates, and new references have been added.*

  - o L725: The applied maximum lifetime of UTLS CO of 90 days seems somewhat on the long side to me. Based not only on Xiao et al. [2007], but also Duncan et al. [JGR, 2007] and Holloway et al. [JGR, 2000], I think that the total atmospheric lifetime of CO over Asian

continental regions in summer is more like 1–2 months, not 2–3 months. So it is possible that the expected range for CO in this figure should be adjusted.

*We agree that the tropospheric lifetime of CO over Asia in summer is probably lower than two months, in agreement with the references that focus on surface and 500 hPa conditions (looking at Fig. 4 in Duncan et al. and Fig. 2 in Holloway et al.) where OH abundances are ~ 2 x $10^6$ molecules $cm^{-3}$ (Fig. 1 in Duncan et al.). At altitudes above the 200 hPa pressure level, i.e. where we observe the CO decline with altitude, OH tends to be significantly lower (below 1 x $10^6$ molecules $cm^{-3}$ in the abovementioned Fig. 1 by Duncan et al.; the vertical gradient in OH is roughly consistent with more recent studies by Naik et al., ACP 2013, and Leliefeld et al., Science, 2018). 1 x $10^6$ molecules $cm^{-3}$ OH corresponds to roughly 70 days CO lifetime, so we believe it likely for the lifetime in the tropopause region is 2-3 months rather than 1-2 months. Nevertheless, we expanded the shaded range to 1 – 3 months and add a small discussion including the suggested references in the main text.*

- L277: Since the fact that no CO mixing ratios exceeding 100 ppb are observed above the LRT is an important point, it might be helpful to add a vertical line on the CO panels in Fig. 4 marking that value.

*We add gridlines to Figures 4 and S7 to make it easier to verify in these Figures all statements with respect to mixing ratios and potential temperature levels.*

- L284: In addition to the point that not all $O_3$ mixing ratios fall in the expected concentration range, it is also worth noting that much of the expected range is unpopulated in the data.

- L286: It is suggested that more rapid ascent may account for the apparently tropospheric character of some of the air masses observed to fall outside of the expected range for $O_3$, but wouldn't stronger ascent also affect the CO in those parcels, leading their measured CO concentrations to extend beyond the expected range (which is not seen)?

*The picture has changed with the new choice of ascent rates, and the discussion is entirely rewritten. This should address both the above comments.*

- L289: Delete "anymore".

*We delete this as suggested.*

- L298: For the reasons mentioned above, I think that this line should be amended to note that 3 months represents the upper end of the range for CO lifetime in the summertime UTLS.

*We delete the "3 months" and now state that we define the lower age limit by the CO lifetime.*

- L301: For clarity, I think it would be good to start this sentence "Considering the upper end of the $N_2O$ range and allowing for a 1 ppb increase between 2016 and 2017, ...".

*We change the sentence as suggested.*

- Fig.5 and its caption:
  - Although I have no objection if the authors prefer to leave Fig. S8 in the supplementary material, I note that there is plenty of room in the observational panel of Fig. 5 for a sizeable inset showing the $O_3$-CO relationship color-coded by $N_2O$. The theta color bar could be moved to the model panel or to the space above the two panels (which would also underscore that it pertains to both). In any case, it would help orient the reader to add the magenta box on the correlation plot in Fig. S8.

*We move Fig S8 to an inset in the left panel of Figure 5.*

- o The grey dashed line in Fig. 5 representing the tropospheric regime is too pale to be easily seen. I think it would be better to make this a black dashed line as well and then add "vertical" and "horizontal" in the caption to distinguish the two black dashed lines.

*We make both dashed lines black as suggested.*

- L315: campaigns --> flights (or campaigns --> campaign)

*We change this as suggested.*

- L325-326: It would be appropriate to add tildes in front of all of these ranges, as is done in each panel of Fig. 6.

*We do this as suggested.*

- Fig. 7: It seems odd to me that the colors used to represent the LRT and CPT have suddenly changed in this figure. It would provide more continuity for the reader to draw the LRT histogram in red and outline the CPT one in blue, in accordance with Figs. 2 and 4. (Of course, the zero line would also need to be colored differently in that case.)

*Colors in the figure are changed as suggested.*

- L328-330: As the authors note, some points lie as much as 45 K above the LRT, thus it is necessary to add "mainly" or some similar qualifier to "are found"; a similar comment can be made regarding the statement about the CPT.

*We add "mainly" twice as suggested.*

- L331-333: The sentence "The points circled ...$NO_x$." feels out of place, tacked on at the end of the paragraph. It might flow better at the end of the previous paragraph. Alternatively, although it is plausible that these points are related to lightning $NO_x$, the authors could do more to back up that statement (e.g., through references to previous studies), in which case this discussion could make up its own short paragraph, probably at the end of the section.

*We move this sentence to the end of the previous paragraph, and we add a more precise localization (the flight on 31 July extended far South, and if the points had been observed at the southern end of the flight track, mixing processes at the anticyclone edge could potentially play a more significant role).*

*In terms of recent convection with enhanced $O_3$ due to fresh lightning $NO_x$, we replace "likely" by "speculatively". Unfortunately, no NOx measurements are available for this flight. A similar incidence of high $O_3$/high CO (together with high NO and low HCl) was observed by Gottschaldt et al. (ACP, 2017, their Figure 6), but they don't explicitly make an attribution in their text.*

L341-345: It is a bit of an understatement to say that the modeled correlation is "somewhat" more compact than that observed–it is considerably more compact, not only in the transition region but also throughout the stratospheric regime. Couldn't the measurement uncertainties (e.g., precision of 20 ppb for CO, 8% for $O_3$) account for some of this scatter?

*We agree that measurement precision is likely to contribute to the spread of the observational data. We rewrite the last two sentences of this paragraph to reflect this.*

- L344: of model's --> of the model's

*We add "the" as suggested.*

**Section 4**

- L351-353: I find this discussion confusing. The authors state that their results provide clear evidence of vertical transport "to the tropopause", but then they go on to note that the data show transport "up to at least 360 and often 370 K", whereas they have shown that the mean

LRT is at 380 K. Moreover, on L374 in the next section, it is stated that convective signatures are occasionally observed up to 380 K. This should be reconciled / clarified.

*We meant to say that 360 K is kind of the lower limit of the convective outflow layer. We changed the statement to "consistently up to 360 K, often 370 K, and occasionally 380 K", which will hopefully clarify this.*

- L354: The wording in this line is awkward. If I have understood it correctly, then rather than "immediate convective outflow" it would be better to say something along the lines of: "convective outflow above the tropopause immediately prior to the measurement time".

*The suggested wording is indeed clearer, so we adopt it.*

- L355: Here too I think the wording is a bit awkward and unclear. I suggest instead: "mixing with the local background following transport to this level, signatures of deep convection".

*We adopt the suggested wording.*

- L358-359: It would be better to add commas after "correlations" and "(Section 3.3)".

*We add commas as suggested.*

- L361-365: I would like to see more discussion here placing these conclusions into the context of other recent studies touching on this issue, including Ploeger et al. [2017], Vogel et al. [2019], Yan et al. [2019], and Legras & Bucci [2020] (and possibly others).

*In this place, we add the Ueyama et al. (2018) reference for the significance of overshooting convection for moistening, and also a reference to Legras and Bucci. For the suggested alternative pathway of slow upwelling, we cross-reference the next Section, where this is discussed in detail and put into context of other work.*

- L371: The year for the Bucci et al. reference should be [2020].

*We change this as suggested.*

- L373: It would be more appropriate to say "the top of the \*main\*convective outflow layer".

*We change this as suggested.*

- Fig. 8 and its caption:
  - Again, it would be nice if the colors of the LRT and CPT overlays matched those in previous figures, even though that would mean choosing a new color palette for the contour plots.

  *Colors are adjusted as suggested.*
  - L758: 30°E --> 30°N

  *We correct this as suggested.*
  - L760: CRT --> CPT

  *We correct this as suggested.*
  - The caption should explain the shading. One possibility would be to begin the sentence about how the zonal and meridional averages are calculated with "As demarcated by the vertical grey lines and bolder colors on the respective panels, ...".

  *We include this in the revised caption.*
- L382: Delete the comma after "(2020)".

*We delete this comma as suggested.*

- L387-388: It would be appropriate to include a reference for the speed of the BDC.

*We add a reference to Wright and Fueglistaler, 2013, here.*

- L388-391: It took me a couple of minutes to figure out that the statement that the observed CO decline and $O_3$ production point to upwelling that is somewhat slower than the ERA-I vertical velocities is based on the fact that the green lines in Fig. 4 were derived assuming an ascent rate of 0.3 K day$^{-1}$< dθ/dt < 0.8 K day$^{-1}$. Since this information appears only in the caption and on the figure itself (not in the main text), it would be good to remind readers of these values here. Moreover, it seems to me that this would also be an appropriate place to remind readers of the uncertainties associated with using PV at 380 K to identify inside-anticyclone points, which is another reason that these upwelling estimates are not "quantitatively conclusive", as noted in L390.

*As stated above, we have changed the 0.3 – 0.8 K day$^{-1}$ range to 0.6 – 1.5 K day$^{-1}$, and the new range is again mentioned here. We also add a statement on the possible impact of the 380 PV criterion.*

- L398-399: As noted in L330-331, Fig. 7 shows the distance above the CPT to be 30 K.

*We change the number from 25 to 30.*

- L399-403: That the tropopause over the ASM region does not represent a vertical transport barrier has been noted in previous studies, e.g., Vogel et al. [2019].

*We add a reference to this study here.*

- L405-409: I find this whole paragraph confusing. For one thing, it's not clear whether the first sentence is meant to be a general statement or an expression of the findings of this study, but in any case it should be made clear that this picture has been described in many previous papers. In addition, I'm not sure why the WACCM simulations reported by Pan et al. [2016] are being discussed here, when simulations with a new (presumably improved) version of the model have been run specifically for this study. Can't these points be made with reference to the simulations shown in Fig. 5 and the animation in the supplement?

*The whole paragraph is rewritten. A reference to Section 3.2 and selected references are added, and reference is made to the new WACCM simulations.*

- L410: Since the analysis here sheds light only on in-mixing of stratospheric air into the anticyclone, whereas most previous studies quantifying the isolation of the anticyclone have focused on parcels escaping from it (and Legras & Bucci [2020] argue that the latter occurs more easily than the former), it might be better to entitle this subsection "In-mixing of stratospheric air".

*We change the Section title as suggested.*

- L413-417: It would be relevant to note here that the analysis of Vogel et al. [2019] showed transport of young air masses in the ASM circulation up to as high as ~460 K.

*We add this reference and modify the discussion accordingly.*

- L417-420: The study of Randel & Park [2006] (already cited elsewhere) should probably also be mentioned here, as they performed a trajectory-based quantification of confinement inside the anticyclone over the range 50–70 hPa.

*We add this reference here as suggested.*

**Conclusions**

- L427: LRT --> LRT around 380 K

*We add this as suggested.*

- L430: times scales consistent with --> times scales largely consistent with

*We add "largely" as suggested.*

- L434: The term "tropospheric bubble" was not italicized in L403; usage should be consistent.

*We consistently use italics in the revised version.*

- L438-440: These lines are somewhat redundant with the two previous bullet points. In addition, the fact that the degree of mixing with surrounding stratospheric air increases at higher levels was noted by Vogel et al. [2019] as well. Finally, the other bullets point to the specific relevant figures, and it would be good if this one did too.

*We want to keep these lines dealing with specifically what happens above 400 K as a separate bullet. But we add a reference to Vogel et al. and also to Figures 4 and 6.*

- L443-444: Unlike Ploeger et al. [2017], Legras & Bucci [2020] found that the "blower" mechanism operates at and above ~360 K, not just above the tropopause. (The studies of Vogel et al. [2019] and Yan et al. [2019] also found substantial horizontal flow out of the anticyclone between 340 and 380 K.) Legras & Bucci further found that localized "chimney"-like behavior ends at the cloud tops, above which ascent follows a broad spiral over the entire anticyclone domain, as shown by Vogel et al. I think it would be beneficial for the authors to put their results into the context of other recent studies in a more detailed and nuanced manner, rather than simply stating that they are "largely consistent".

*This discussion is expanded and made more specific in the revised version.*

- L445-449: The importance of extreme deep convective clouds in moistening the ASM region is also emphasized by Ueyama et al. [2018], and Legras & Bucci [2020] also found that penetrative convection may have a significant impact.

*These references are added to the discussion of deep convection.*

**References**

- L490: The paper by Bucci et al. has now been accepted and is in press.
- L570: The paper by Legras & Bucci has now been published.

*We update all references upon resubmission and also expect to get a final update with copy-editing*

- L670: the Comission--> the Commission

*We correct this as suggested.*

**Supplementary Material**

- L771: CRT --> CPT

*We correct this as suggested.*

- L789: on --> in

*We correct this as suggested.*

- L803: Figure 3 --> Figure 4

*We correct this as suggested.*

---

## Author Comment (AC2) · 15 Dec 2020

*We thank the reviewer for the constructive criticism and suggestions that significantly improve the paper. Below, we go through them point by point, using* normal font for original remarks *and* blue italics *for our replies.*

The manuscript endeavors to comprehend the field measurements of CO, $N_2O$ & $O_3$ from the ASMA region during the StratoClim field campaigns and elucidate its different transport pathways. Quite a lot of quantification and new insights are offered in this manuscript. Hence the manuscript can be accepted for publication after addressing the following points.

Recommendation: Minor revision

Comments/suggestions

1. 3D Map of averaged CO from figure 2, looks nice but hard to get information easily. The Theta_e labels are not visible and difficult to understanding CO mixing ratios difference as they mentioned in the manuscript.

*Figure 2 is completely refurbished, and the use of an incorrect color pallete in the displayed data is corrected (cf. reply to Michelle Santee's comment).*

2. Many unconventional acronyms used in the manuscript, which make it difficult to read the manuscript, and some of them are not expanded where it first appeared. It will be better to minimize the same in running text.

*We have carefully checked this and spell out acronyms the first time they are used in the revised manuscript.*

3. Hope that AM Eq latitude in the figure 3 label is a typo instead of M Eq Lat. Or AsianMonsoon Equalent latitude is also a better terminology.

*The use of AM EqLat in Figure 3 was unintentional and is corrected to MeqLat.*

4. In case if not much required the supplementary materials can be limited in the manuscript. For eg Figure S4, which is not providing any new insights and besides which creating confusion with the discussion later in the text and main figures.

*We prefer to keep all supplementary figures. They are in the supplement, because they are not strictly necessary to understand the paper or for the line of arguments. But they provide additional information that some may find interesting. For example, the rational for a PV based boundary of the ASM anticyclone and the derivation of MeqLat as a new coordinate was described by Ploeger et al. (2015) for the monsoon season in one particular year (2011), and the two analogous (to Fig 13 in Ploeger et al., 2015) plots for 2016 and 2017 shown in Figure S4 to give a better feeling for what the same analysis looks like in the StratoClim years when looking at the entire monsoon season. And it may help to understand why slightly different PV threshold numbers are being used for these years.*

5. The statements in lines 245's are very difficult to digest from the figure and considering the possibilities of multiple other influencing factors.

*We see the strong change of LRT altitude with latitude in the reanalysis data, consistent with the strongly changing difference between CPT and LRT at these latitudes seen in Figure S3 and with the strong PV gradient at 380 K seen in Figure 1. We add this information in the revised version.*

6. Why there is a high value of CO even below 340K at designated latitudes (from figure 3 & 4). It will be nice if the authors can provide some details/explanations in this regard.

*We are not sure what observation exactly this comment addresses.*

*In Kathmandu, tropospheric CO is consistently high (~100 ppb with some higher values especially at low altitude), which is expected and discussed. In Kalamata, tropospheric CO is generally lower, but some high values related to local pollution at Kathmandu airport show up at the lowest altitudes (especially in figure 2).*

7. In the discussion section how suddenly authors restricted the transport till 370K. Till this point, the authors were mentioning that the LRT is at 380K. Better to clarify this.

*We separate the terms "troposphere" and the potential temperature layers into two separate statements, which will hopefully clarify this. Also see our response to a similar comment by Michelle Santee.*